# Advanced Resources Reservation in Mobile Cellular Networks: Static vs. Dynamic Approaches under Vehicular Mobility Model

Peppino Fazio [1,2,*] and Mauro Tropea [3]

1 Department of Molecular Science and Nanosystems (DSMN), University Ca' Foscari, Via Torino 155, Mestre, 30170 Venezia, Italy

2 Vysoka Skola Banska, Technical University of Ostrava, 17. Listopadu 2172/15, 708 00 Ostrava-Poruba, Czech Republic

3 Department of Informatics Engineering, Modellistics, Electronics and Systems (DIMES), University of Calabria, Arcavacata di Rende, 87036 Rende, Italy; mtropea@dimes.unical.it

* Correspondence: peppino.fazio@unive.it

**Abstract:** Many studies in literature have shown that the bandwidth of an ongoing flow can dynamically change during multimedia sessions and an efficient bandwidth allocation scheme must be employed. This paper focuses its attention on the management of predictive services in Wireless Infrastructure Dynamic Networks. In particular, two classes of service are considered: NSIS-Mobility Independent Predictive and NSIS-Mobility Dependent Predictive, where NSIS is the Next Steps in Signaling protocol, employed for resources reservation in Integrated Services architectures. A general prediction technique is proposed, based both on the analysis of time spent into a cell by mobile nodes and on the probabilities of hand-in and hand-out events of mobile nodes from wireless cells. User mobility needs to be firstly analyzed and a novel realistic mobility model has been considered, differently from some existing works in which synthetic mobility is generated. The analysis of user mobility is mandatory when the reduction of passive resource reservations for NSIS-MIP users is desired, with a good enhancement in system utilization. Moreover, predictive reservation and admission control schemes have been integrated. The performance of the 2D wireless system is evaluated in terms of average system utilization, system outage probability, number of admitted flows and reservation prediction errors. We provided to carry out an extensive simulation campaign, in order to assess the goodness of the proposed idea: we verified that good results (in terms of perceived utility, bandwidth and admitted flows) can be obtained, outperforming also some existing works.

**Keywords:** CAC; bandwidth allocation; utility function; Markov model; path prediction; wireless networks

## 1. Introduction

In the last 20 years, the demand for wireless devices and applications has growth greatly, because of the manifold advantages offered by these telecommunication systems. User mobility is the main reason for received Quality of Service (QoS) fluctuations, because it heavily impacts on the desired QoS parameters. During the hand-over events, the available bandwidth in the new cell (which can be a Base Station BS or an Access Point AP, depending on the considered technology) may be scarce, while the perceived service quality may go outside the requested bounds. In addition, the current active call may be disconnected from the network. It is clear that deterministic services (as used in wired networks, where the QoS fluctuations are negligible) become inadequate in wireless scenarios and more flexibility is needed, allowing elastic and variable QoS. Using the adaptive networking paradigm permits the mitigation of the so-called "bandwidth availability" in dynamic networks, where users can require different levels of QoS.

Wireless networking introduces several issues regarding QoS (such as limited bandwidth and high error rates), due to fading and mobility [1,2] effects. A suitable approach

introduced in [3] was used, based on the Finite State Markov Chain (FSMC) for channel modeling, accounting for multipath fading effects in terms of BER and channel capacity, directly related to the different modulation schemes, Doppler shift effects and user average speed. When performing resources reallocations, in addition to channel conditions, instantaneous user satisfaction level must be also taken into account, in order to introduce a certain level of fairness, between flows that belong to the same service class.

This purpose is achieved by introducing utility functions [4], as a way to describe user satisfaction profiles based on the instantaneous quality of the perceived service. We base our proposal on an architecture able of reserving different bandwidth levels while, at the same time, offering guaranteed services [5–9]. In particular, we provided to integrate Integrated Services Packet Networks (ISPNs), the Mobile Resource ReSerVation Protocol (MRSVP) and the Next Step In Signaling (NSIS) [5,6] for managing users mobility and exchanging state information of the wireless network and the single connection reservations.

In addition, routing protocols have great importance in the mobility in wireless network and new paradigms and technologies are used for managing the energy issues in this context such as swarm intelligence-based approaches [10]. We propose also a novel Call Admission Control (CAC) and a Bandwidth Reallocation Scheme (BRS). In this work, in particular, two classes of service were considered: NSIS-MIP and NSIS-MDP [6]. Given the high impact of the mobility model on the obtained results, we considered a realistic mobility generation for our simulations, because the obtained results can be unrealistic if the model is not adequate. There are different proposed models for describing users' behaviors, such as [11–14]: unfortunately, all of them take into account a synthetic approach, based on analytical and/or stochastic equations, able to describe user movements in different environments (urban, rural, etc.), without considering the existence of real topologies and road structures. In this work, instead, the Citymob4Roadmaps (C4R) proposed in [15,16] has been considered, with realistic behaviors, because mobility traces are generated according to roads structures, extracted from concrete environments. By analyzing user mobility, a prediction technique is proposed for NSIS-MIP flows, in order to evaluate the cells that a user with an active flow will probably visit, achieving a higher system utilization. The proposed idea is based on two important statistics: the Cell Stay Time (CST) probability distribution and the Hand-off Directions Probabilities values (HDP).

The main contributions of this work are:

(1) Extension of MRSVP protocol with additional features inherited by NISIS, in order to dynamically manage NSIS-MIP and NSIS-MDP classes also during on-going calls;

(2) Proposal of a utility-based rate adaptation scheme with the application of utility functions for the management of Best Effort (BE) associated to NSIS-MDP class and video traffic associated to NSIS-MIP class;

(3) Proposal of a Hand-off Direction and CST based reservation schemes in order to meet adaptive QoS requirements during users' movements. In-advance bandwidth reservations is proposed in order to reduce the QoS degradation and call dropping probability;

(4) Static and Dynamic (threshold-based) reservation prediction schemes proposal considering real mobility traces;

(5) Validation of the considered scheme through realistic mobility patterns.

This paper is organized as follows: Section 2 gives an overview of the related work concerning the proposed reservation techniques and studies in wireless environment. Section 3 describes the signaling protocols adopted to make the reservation and the supported QoS classes. The considered mobility generation scheme has been briefly introduced in Section 4; the utility functions adopted in the call admission control and bandwidth management are presented in Section 5; the proposed static and dynamic bandwidth reservation schemes are presented in Section 6. In Sections 7 and 8 simulation results and conclusions are, respectively, summarized.

## 2. Related Work

The resource reservation mechanisms, such as resource ReSerVation Protocol (RSVP) [17,18] and packet scheduling algorithms, have been extensively used to satisfy applications QoS requests. However, as described in the previous section, mobile computing requires elastic mechanisms, such as Rate Adaptation mechanisms [19], able to satisfy dynamic and degraded requests. In ISPNs, each single flow can receive a QoS level which can be negotiated by the RSVP in static scenarios [17,18], or MRSVP and Dynamic ReSerVation Protocol (DRSVP) in mobile networks [20–22]. We considered the NSIS [23,24] in order to take into account its capability of establish, maintain and remove control states in network nodes. The new protocol proposed by IETF can be viewed as a new extensible IP signaling architecture. In the following we present two sections related to literature contribution to reservation scheme and mobility applications.

### 2.1. Reservation Schemes and Prediction in Literature

In this sub-section we will introduce some contribution of scientific research about mobility prediction and reservation scheme topics.

In [12], a scheme for the service patterns estimation by tracking mobile users is proposed: it is based on historical records and permits the estimation of the future cells which will be probably visited by a mobile user.

The model presented in [25], instead, does not exploit the knowledge of the users' mobility history. In this way, the model is limited to make a prediction for the part of the network (cells, geographical area, etc.) where mobile users are likely to move (i.e., user's final destination), instead of the complete path to reach the final destination.

In [26] the authors propose a mobility prediction scheme, based on Dempster-Shafer processes and Markov theory. In particular they consider a typical transportation topology, composed by roads and intersections, having the possibility of dividing the considered region into cells. They assume that mobility history can be analyzed periodically, mining information about user habits. The proposed DAMP scheme stores time and geolocation coordinates, and it is shown that the whole process can be modeled as a semi-Markovian one, obtaining high accuracy values (in terms of roads similarities). DAMP is able to predict mobility positions within a predefined time period.

In [13] a mobility prediction scheme for cellular networks is proposed, in order to optimize hand-over procedures: the main idea is to a-priori know hand-off times, for speeding up hand-over switching. Mobility analysis is performed by a Kalman filter (for tracking GPS signals) and Hidden Markov Models (HMMs—for probability analysis). Both results are used for the estimation of the next probable cell.

In [27], the location prediction using mobile phone traces has been considered. The authors made a deep analysis on mobility patterns, concluding that they are strongly correlated with co-located patterns and they affect user short-time mobility. They proposed a new prediction scheme, based also on the social interplay revealed in the cellular calls.

In [28] the authors show how the knowledge about future mobility locations can enhance system performance in terms of hand-over droppings and blocking probabilities. The proposed MPBR is able to evaluate hand-off times (with a detailed traffic analysis) and future available bandwidth, integrating them with a CAC scheme. The obtained results show that service continuity can be guaranteed, while maintaining a good level of system utilization.

The work in [29], argues about the management of Mobility Independent Predictive (MIP) and Mobility Dependent Predictive (MDP) services in adaptive networking. Adaptive QoS (also indicated as soft QoS) is the main concept of the paper, able to increase the overall wireless system utilization, on the base of a utility-based BRS.

The works in [30,31] propose some prediction techniques based on the CST distribution under the Random Way Point (RMM) model, arriving at a closed form for binding average speed, cell coverage radius and variation of the average speed. Moreover, in [32,33] an unconventional-based prediction scheme (DPBMA) has been proposed, in order to man-

age the concept of passive reservations. A multiplexing algorithm has been integrated, in order to avoid wasting a lot of passive reservations. A distributed set of Markov chains has been adopted, evaluating its performance in terms of prediction accuracy, through many deep campaigns of simulations.

### 2.2. Mobility Applications in Literature

Many works regarding mobility management and applications have been proposed in literature, so we will briefly describe some of them.

In [34] the importance of mobility management is considered, by taking into account several monitoring issues (large-scale monitoring, search and rescue tasks, pursuit and escape operations, etc.). The authors propose a novel algorithm to improve the monitoring efficiency of mobile wireless nodes, considering the reduction of the overlapping rate of the involved wireless nodes (avoiding wastage of covered area).

In [35], another interesting application of mobility management is described: the authors propose a new algorithm able to guarantee a periodical cooperative sweep of mobile sensors, in order to cover all the desired Point of Interests (PoIs) of a given surveillance region. The novelty of the proposed idea consists in the cooperation of sensor nodes, by deploying them on the same trajectory, in order to reduce the sweep time.

The authors of [36] propose a new optimization algorithm, able to manage nodes mobility with the main aim of reducing the total energy consumption, while maintaining the optimal coverage of the considered area. Their idea is based both on Fuzzy Logic (a Fuzzy Inference System governs sensor movements) and Swarm Intelligence and it is able to achieve the minimum energy consumption while optimizing sensors movements, in order to cover the maximum achievable area.

In [37,38] the authors underline the importance of having real mobility traces (naturalistic mobility) instead of synthetic ones, as well as the difficulty to retrieve mobility logs extracted from general scenario, not constrained to specific situations, events and contexts. In order to solve this impractical issue and to have the possibility to collect a huge set of data, useful for research activity, the authors propose to map naturalistic driving information with the Geographic Information Systems (GIS). In this way it is possible to arrange a very deep understanding of human driving behavior, avoiding the drawbacks of the traditional methodologies and measuring a great number of parameters at high temporal frequencies, having the possibility to manage also the relative and absolute errors, which can occur and can be located by the cartographic representation of GIS.

## 3. The Considered Architecture

This section gives a comprehensive overview of the considered architecture and protocols, in order to have a clear idea about the concept of passive reservations.

### 3.1. Protocols and Service Classes

As mentioned in Section 2, in ISPNs, all flows can receive different QoS, which must be obtained when the sessions start, in order to verify if the QoS-aware relay nodes are able to support them. In particular, the MRSVP active/passive policy (described in the following) and the MRSVP service classes are used, while the soft state signaling approach [17,23] is the same of the NSIS scheme, because in the network nodes the "non-permanent" control state expires unless refreshed, while the considered level of bandwidth may vary between a minimum and a maximum level.

In this way, NSIS Signaling Layer Protocols (NSLPs) can signal for any QoS model (as the IntServ). In addition, MRSVP uses UDP for transport mechanism and discovery: signaling message delivery are combined into a single protocol step (it does not provide a solid security framework). Instead, all NSIS nodes necessarily do not support all signaling applications, because signaling message transport and signaling applications are strictly separated (by the NSIS Transport Layer Protocol—NTLP and NSIS Signaling Layer Protocols—NSLPs). Decoupling of discovery and transport of signaling messages is re-

alized in NSIS through the introduction of a session identifier. For details see [39]. We based our work on the service classes provided in [5,6] and, as described in Section 1, the QoS state is maintained through the NSIS protocol, so the prefix NSIS is used in order to identify the presence of the NSIS protocol for signaling operations. In [5,6], three service classes are provided:

(a) NSIS Mobility Independent Guaranteed (NSIS-MIG, which provides intolerant applications, with very stringent guarantees on packet delays and jitter);

(b) NSIS Mobility Independent Predictive (NSIS-MIP, dedicated to elastic real-time applications, able to work with some resource bounds);

(c) NSIS Mobility Dependent Predictive (NSIS-MDP, which can be compared with the best-effort class, subject to continuous QoS degradations and/or droppings).

*3.2. Advanced Resource Reservations with NSIS*

In our work, we deal with NSIS-MDP and NSIS-MIP: NSIS-MIP services are admitted into the system if a pre-reservation phase on the future visited cells obtains a positive answer from the system (passive reservations). NSIS-MDP services, instead, do not provide passive reservations and can reserve the bandwidth only on the active cell (where the call has originated). For both NSIS-MIP and MDP, adaptive bandwidth reservation is admitted [20,31], so the assigned resources can change during the active session.

Obviously, this kind of behavior during the Call Holding Time (CHT) can guarantee a flexible resource management (not possible for NSIS-MIG), with a huge increase of system utilization. Talukdar et al. proposed the MRSVP [5] to support mobility for RSVP. The main advantage of the MRSVP is the ability to make active and passive reservations [5], as discussed in the following. Our attention is neither focused on the signaling performance of the employed protocol nor on the management of hand-off events in terms of protocol signaling activities, but it is important to consider how a user can make a passive reservation on a remote location [30].

The most important concept in MRSVP is the set of *proxy agents*; there are two types of proxy agents:

(a) local proxy agents, generally identified as the active cell in which a mobile host is making the service request (hence, the active reservation); NSIS-MDP users make use of local proxy agents only (they request the service only on the current cells);

(b) remote proxy agents, generally identified as the passive cells which will be probably visited by mobile hosts (hence, dealing with passive reservations); NSIS-MIP make use of remote proxy agents in order to manage their passive requests remotely.

The scenario considered in this paper consists of the following elements: each cell is covered by an AP, which is connects to the Internet by a Switching Subnet (SS). The active reservation request (made by mobile hosts to the current cells) is routed to the sender node by appropriate NSIS messages. For the NSIS-MIP class, the Pre-RESERVE message is sent to the local AP (and to the remote ones through the SS); all the involved APs will answer with a positive acknowledgement (ack) if enough bandwidth is available.

If the NSIS-MIP user will not receive all the positive acks (RESPONSE NSIS messages), then it will retry to connect later. If all the APs (active and passive) will have the possibility to accommodate the new request, the mobile NSIS-MIP will perform its reservation request by sending the RESERVE message (active-RESERVE for the local AP, passive-RESERVE for the remote APs). If the requesting node belongs to NSIS-MDP class, messages are exchanged only with the current active AP (for details see [39]).

## 4. CityMob and C4R-GUI for Generating Real Mobility Traces

In our work, among different solutions [11–14] for mobility generation, we considered CityMob and C4R v1.0 from GRC of the UPV of Valencia [15], given its capability of describing real users' movements. It is a mobility pattern generator designed to investigate different vehicular mobility models, and their impact on inter-vehicle communications. It is able to create urban mobility scenarios, with damaged cars, traffic lights and downtowns.

It features three different sub-models (Simple Model, Manhattan and Downtown), combining a certain level of randomness and giving the opportunity of considering the Open Street Map (OSM) [40] database: it is an open web service, offering data distributed with the Open Data Commons Open Database License (ODbL) by the OpenStreetMap Foundation (OSMF). In addition, C4R gives the possibility of importing a desired map in SUMO format [41] and applying the speed/acceleration models as proposed in literature [42,43] (Figure 1). The strength of C4R consists in the possibility of taking into account human behavioral features applied to real roads and movement segments, by extending the differential equation:

$$\frac{dv}{dt} = F_1(d, v, V) + F_2(d, v, V)\varepsilon \tag{1}$$

where $d$ is the distance to the in-front vehicle (from one bump to the other one), $v$ is the speed of the considered car, $V$ is the speed of the leading car and $\varepsilon$ is a noise term, while $F_1$ and $F_2$ describe the reaction of the driver to the in-front situation. In fact, more recent studies [43] consider:

$$b(v + a^*\tau) + v\tau + \frac{1}{2}a^*\tau^2 \leq b(V) + d \tag{2}$$

where $b$ represents the braking distance function, $a^*$ is the optimal acceleration, $\tau$ is the considered time horizon. By the studies in [42,43], it has been shown that:

$$a* = -\frac{v}{\tau} - \frac{b}{2} + \sqrt{\left(\frac{v}{\tau} - \frac{b}{2}\right) + \frac{2bd + V^2 - v^2}{\tau^2}} \tag{3}$$

should be satisfied if a realistic model has to be considered. This is one of the most important feature of CityMob (and C4R), which leads to obtain realistic traces of mobile vehicles inside the map.

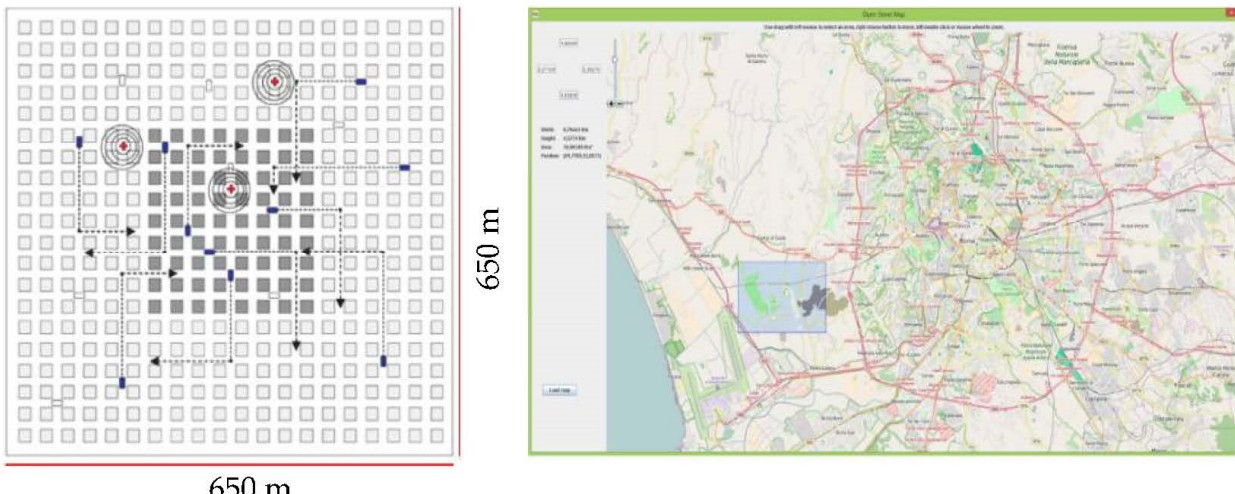

650 m

650 m

**Figure 1.** Examples of CityMob Manhattan sub-model (on the **left**) and C4R interface on the **right**.

## 5. Traffic Models and Utility Functions

As introduced earlier, in our work the CAC and the BRS are carried out by a utility-oriented algorithm, able to take into consideration the time-varying nature of wireless links, between hosts and APs [3]. Service requests can be made by NSIS-MIP or NSIS-MDP mobile users and bandwidth is allocated to each of them (in the case of a positive answer from CAC) by respecting a lower bound on the perceived instant utility. The algorithm is able to guarantee, also, that in the long-run the bandwidth is fairly allocated and efficiently utilized [44]. The considered traffic profiles, associated to NSIS-MDP and NSIS-MIP users, are Best Effort (BE) and Real Time Video (RTV), respectively.

### 5.1. Real Time Video Traffic for Mobility Independent Predictive Services

When a real-time session is initiated, all the involved applications are extremely sensible to the packet delivery delay and they become unsuitable if the delay overcomes some fixed bounds. The needed data rate during a real-time transmission must be almost constant so, depending on data transmission and compression techniques, a minimum and a maximum level of employed bandwidth can be determined. If utility functions are associated to real-time traffic, the principle of utility max-min fairness must be respected [4,45] (when the perceived utility of a user is increased, the utility of another user, which receives already a smaller utility, is decreased. In this paper, we consider multi-layer video streaming as real-time traffic for NSIS-MIP users [4]. The satisfaction levels for a multi-layer streaming content can be effectively described by a multi-step utility function, where each step corresponds to an encoding layer, representing the achieved utility at that layer. So the general considered expression for an n-layered compressed video is [4]:

$$U_{RT}(r_s) = \frac{a_{s,k}}{1 + \exp(b_{s,k}(r_s - c_{s,k}))^{d_{s,k}}} + \sum_{j=1}^{k-1} a_{s,j} \tag{4}$$

where $r_s$ is the received rate level, $a_{s,k}$, $b_{s,k}$, $c_{s,k}$, $d_{s,k}$ are normalization factors and $c_{s,k} - \alpha_{s,k} \le x_s < c_{s,k+1} - \alpha_{s,k+1}$, for $k = 1, \dots, n$, $\alpha_{s,1} = c_{s,1}$, $\alpha_{s,k} = \frac{c_{s,k} - c_{s,k-1}}{2} > 0$ for $k = 2, \dots, n$. Figure 2a shows a multi-step utility function for the considered traffic, with the same values used in [4], except for: $a_{s,1} = 1$, $a_{s,2} = 1.7$, $a_{s,3} = 0.9$, $a_{s,4} = 1$, $c_{s,1} = 512$, $c_{s,2} = 640$, $c_{s,3} = 768$, $c_{s,4} = 896$.

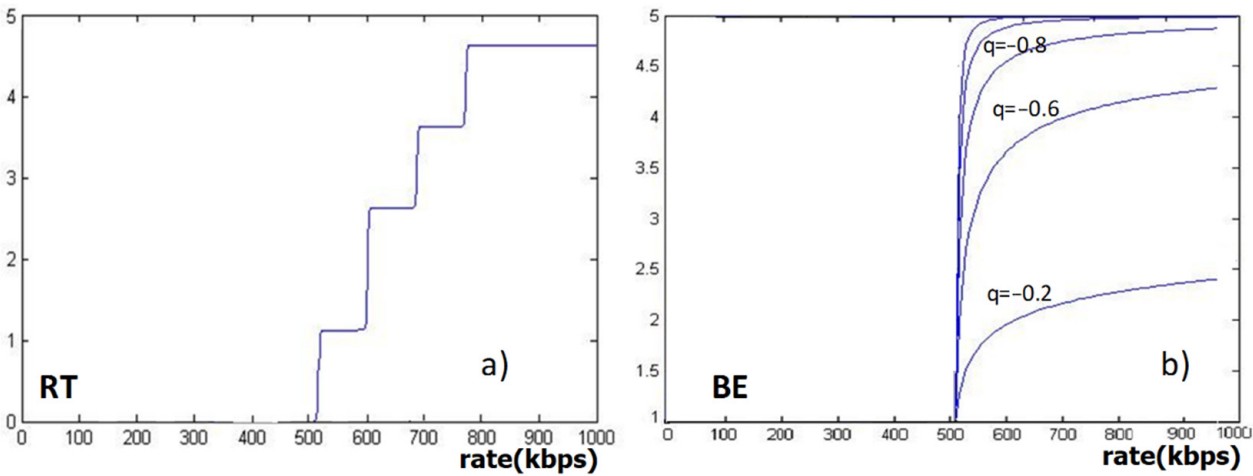

**Figure 2.** Utility functions for Real Time (**a**) and Best Effort (**b**) traffic.

### 5.2. Best Effort Traffic for Mobility Dependent Predictive Services

As earlier introduced, the BE traffic has been associated with NSIS-MDP users, because their services cannot be independent from users mobility. An important characteristic of BE traffic is represented by its capability to adapt to the available bandwidth: all the applications that belong to BE traffic are able to use the entire available bandwidth during data transfers, always respecting the existing upper bound (no guarantees are given for the lower bound, so a flow may be easily dropped).

In this way, transmission speeds are highly time variant. Utility functions for elastic applications are concave over the entire range of bit-rate, approaching their maximum only when the perceived rate tend to infinite [45]. Utility curves can be implemented for any given network performance matrices, such as rate and delay. The most used way for building-up utility curves is based on subjective surveys: users are asked to give an opinion

on the application performance under a wide range of network conditions. We based our choice on the surveys conducted in [45], so an exponential function can be considered:

$$U_{BE}(r_s) = a + b \cdot r_s^q \tag{5}$$

where $q < 0$ and $b < 0$, because $U_{BE}$ must be a monotonically increasing function of $r_s$; $a$ is the maximum desired utility value and when $r_s$ approaches to infinity, $U_{BE}(r_s)$ approaches to $a$; from [45], generally $a = 5$. Figure 2b illustrates the trend of $U_{BE}$ for different values of $q$ with $a = 5$ and $b = -9$.

### 5.3. Utility-Based Bandwidth Management

The proposed idea takes into account the channel link quality, modeled as described in [3,31,46,47] and aims to guarantee a good level of fairness for users belonging to the same service class. Another aim of our proposal is aimed at obtaining a very low outage probability $p_{outage}$ (that is the probability associated to the event that instant perceived utility of any served user falls below its lower threshold). As described in [3,31,46], the wireless link among a user and its current active AP can be modeled by a k-state Markov chain: the average state-permanence time associated to each state $m$ is indicated with $t_m$, the degradation ratio of the $m$-th state is $D_m$ (with $0 \leq D_m < 1, \forall\ 1 \leq m \leq k$). If $r_i$ is the amount of bandwidth allocated by the network to user $i$, the received instant utility is defined on the basis of the belonging service class:

$$u_i = U_i - (1 - D_i, m) * r_i) \tag{6}$$

while for the specific service class it will be:

$$u_{MIP,i} = U_{RT}((1 - D_{i,m}) * r_i) = \frac{a_{s,k}}{1 + \exp(b_{s,k}((1 - D_{i,m}) * r_i - c_{s,k}))^{d_{s,k}}} + \sum_{j=1}^{k-1} a_{s,j} \tag{7}$$

$$u_{MDP,i} = U_{BE}((1 - D_{i,m}) * r_i) = a + b \cdot [(1 - D_{i,m}) * r_i]^q \tag{8}$$

It must be outlined that $D_{i,m}$ is associated to every chain state when the FSMC for uIer $i$ is defined, so it is a fixed value. For more details about the FSMC, to see [3,31,46]. The implementation of the wireless channel has been carried out according with the standard IEEE 802.11 [46]. In order to guarantee the intra-class fairness level, a normalized gap of the average received utility can be defined for user $i$ as:

$$G_i = \frac{(u_{i,avg} - u_{CLASSi,min})}{u_{CLASSi,min}} \tag{9}$$

where $u_{i,avg}$ is the average received utility by user $i$ and $u_{CLASSi,min}$ is the minimum allowed utility level foI user $i$ that belongs to the CLASS service class (NSIS-MIP or NSIS-MDP). The proposed algorithm follows an intra-class fairness criterion, so each value of $G_i$ will be compared only with $G_j$, where users $i,j$ belong to the same service class. Figure 3 represents the main phases of the proposed reallocation algorithm.

When a link degradation occurs to the $i$-th used, it will give-up some bandwidth to another user $j$, with a smaller normalized gap. When the quality of $i$'s link becomes better, it will receive some bandwidth from user $j$, with a larger normalized gap. In this way, the combined instant utility will be maximized. For example, in order to satisfy user $i$'I $u_{CLASS\ i,min}$ when its link degrades and to avoid the outage event, the scheme searches for *CLASS-benefactor(s)* (able to give-up some bandwidth), starting from the user with the largest normalized gap. A deep and complete description of the upgrade/degrade effects on bandwidth reallocations can be found in [31].

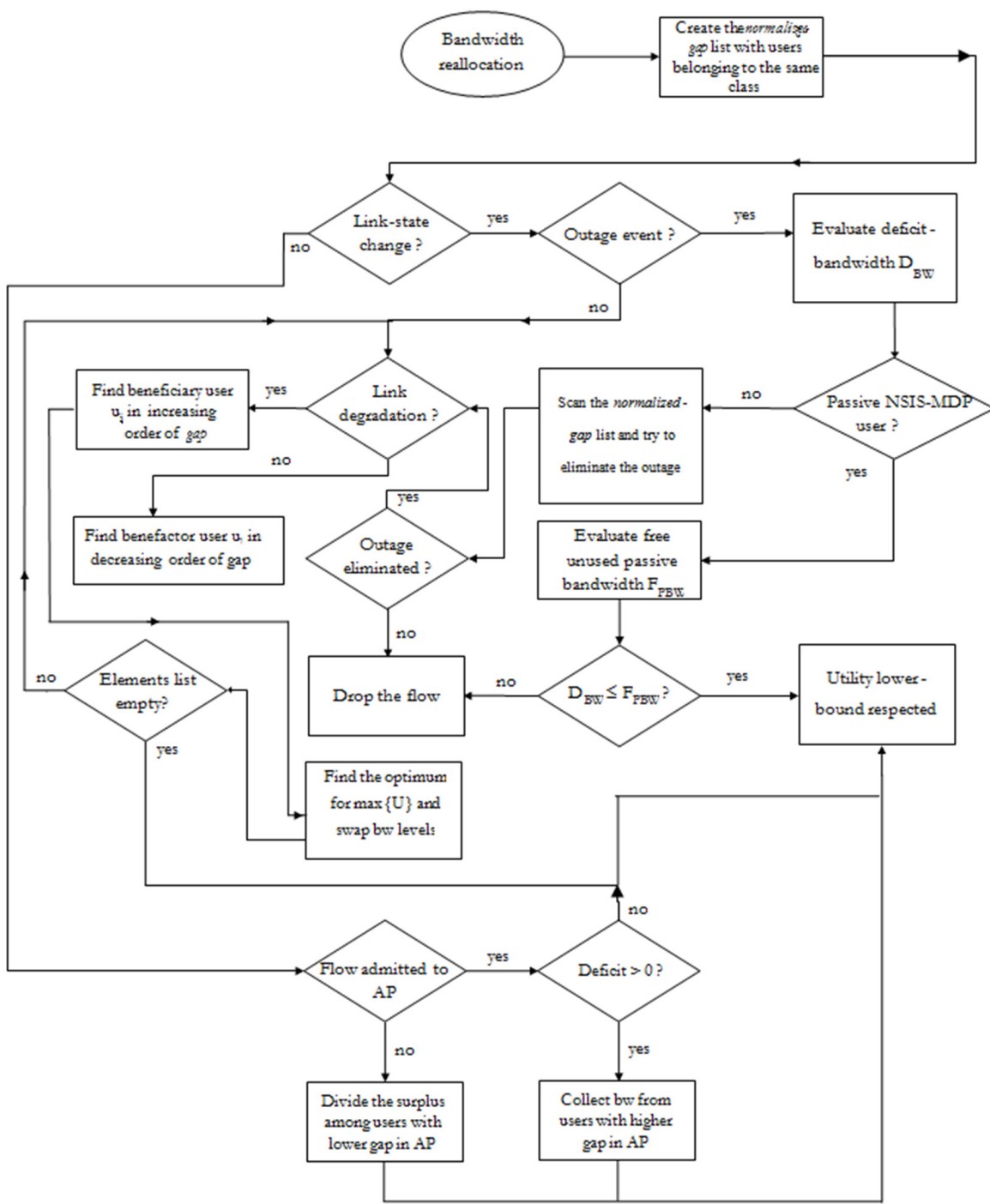

**Figure 3.** Bandwidth reallocation algorithm state flow diagram.

In addition, if a user arrives (new user) or departs from the network, the BRS has to take place. For example, if, after a new user arrival, $r$ bandwidth needs to be collected (from current *CLASS*-users), user $j$ with largest normalized gap $G_j$ will give up an amount of bandwidth equal to $\min(\max(0, r_j - \frac{r_{j,\min}}{1-D_{j,q}}), r)$, where $q$ is $j$'s link state. The procedure is carried-on until $r$ has been reached (or all the current users have been searched). If $r$ is not reached, a second collection round is started but, this time, each chosen user will be dropped out from the network. A user with a totally degraded communication link has to give up a part of its reserved bandwidth in order to maximize the perceived utility of system users (for details, see [31]). As will be seen in next section, an NSIS-MDP user can use free available bandwidth in the current AP (active-NSIS-MDP) or a certain amount of

passive bandwidth that is reserved for NSIS-MIP flows that will come in the current AP (passive-NSIS-MDP).

As regards the time complexity of the proposed scheme, assuming there are *n* admitted users, an update of the normalized gap must be made, with a complexity of O(n), then the AP has to sort the users list (it can be assumed to have a complexity of O(nlogn)). When a BRS operation needs to be made, the benefactor and the beneficiary are found with a single list scan in O(n) time. Comparing the complexity terms, we can conclude that the algorithm performs with a time complexity of O(nlogn) in the "worst case".

The CAC algorithm is different for the two classes of service (NSIS-MIP, NSIS-MDP). For NSIS-MIP class, the flow is admitted if:

$$\sum_{c=1}^{C} p_{0-MIP,c} \leq C \cdot p_{outage-MIP} \tag{10}$$

where *C* is the number of cells that mobile host will visit (that will be determined through a dedicated algorithm as in the next sections) and $p_{outage-NSIS-MIP}$ is the outage probability of the wireless system. As regards NSIS-MDP users, the condition in Equation (10) is verified only for the current cell. When a new user makes a service request, the CAC calculates $p_{0,c}$ and verifies if it is less or equal $p_{outage}$ for each cell: in this way the new request can be admitted or rejected (if it is not satisfied). The outage probability $p_0$ for user *i* belonging to the *CLASS* service class at any time is:

$$p_{0-CLASS} = \Pr\left\{\sum_{i=1}^{n} \frac{r_{i,\min}}{1 - D_{i,m_i}} > R_{CLASS}\right\} \tag{11}$$

where $m_i$ is user *i*'s link state, $p_{mi}$ is the probability of the user *i*'s link to being in state *m*, *n* is the total number users (with the new one) belonging to *CLASS* and $R_{CLASS}$ represents the bandwidth dedicated to *CLASS* services. For a deeper analysis of the proposed CAC and channel model with degradation states to see [30].

## 6. Resource Reservation Schemes: Static vs. Dynamic

The considered system consists of a set of wireless clusters, as illustrated in Figure 4. The Call Arrival Time (CAT) follows a Poisson distribution, while the *CHT* is exponentially distributed. In order to evaluate the number of cells $C_e$ visited by the mobile host during its *CHT*, the *CST* of mobile hosts has been obtained by a preliminary set of simulations: it follows a Gaussian distribution under the CityMob generator, with different values of maximum acceleration and maximum speed [29,31]. $C_e$ can be evaluated as in [31], but without any information about user mobility pattern (directional preferences), $C_e$ can be used only to make circular reservations (Figure 4), around the active cell. In this way, following the same approach of [48,49], the number of required passive reservations $C_r$ for NSIS-MIP services increases with polynomial trend, as indicated in Equation (12). Table 1 shows the numerical values for Equation (12) and for the directional reservation policy P, shown later:

$$C_r = 3 \times C_e \times (C_e - 1). \tag{12}$$

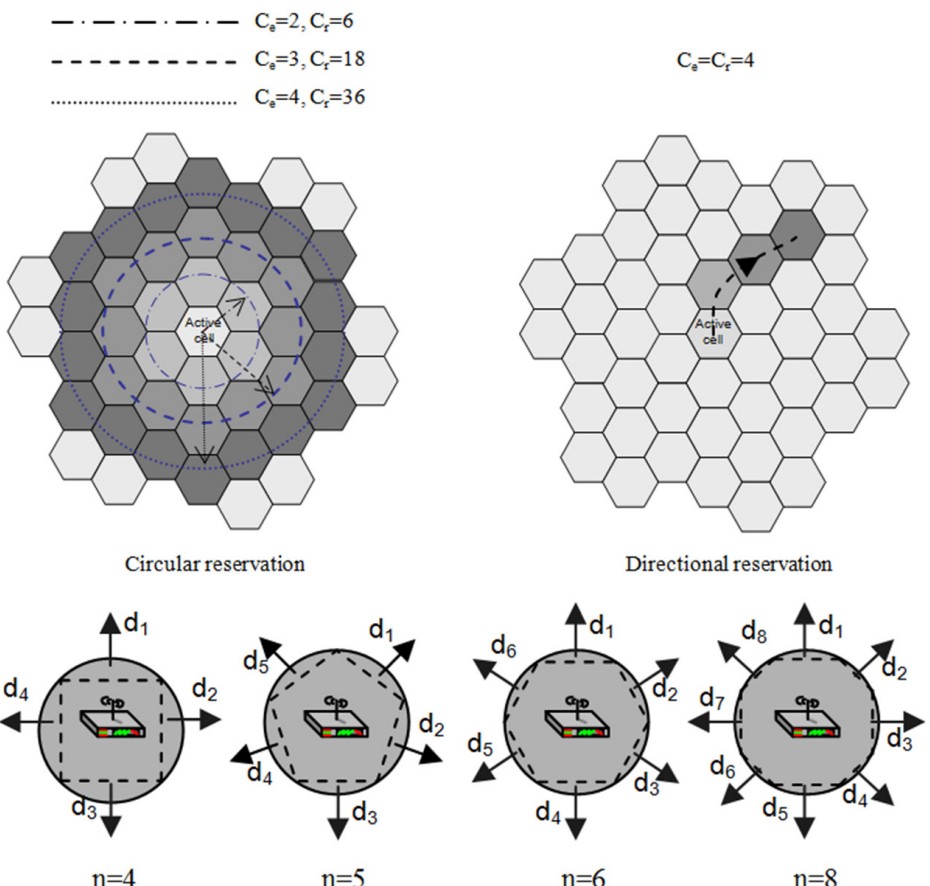

**Figure 4.** Simulated topology with circular/directional reservation policies (hexagonal coverage).

**Table 1.** Number of cells involved in the bandwidth reservation phase, for different policies.

| Mobility Parameters | P(1-1-1) | | P(1-2-3) | | P(3-3-3) | |
|---|---|---|---|---|---|---|
| | Cr(P) | Cr | Cr(P) | Cr | Cr(P) | Cr |
| $a_{max} = 1.4$ m/s$^2$, $a_{min} = -2$ m/s$^2$, $\tau = 0.3$ $\mu = 29.26$ s, $\sigma = 0.45729$ s | 5 | 60 | 22 | 60 | 52 | 60 |
| $a_{max} = 1$ m/s$^2$, $a_{min} = -2.5$ m/s$^2$, $\tau = 0.4$ $\mu = 62.22$ s, $\sigma = 5.09547$ s | 2 | 6 | 3 | 6 | 4 | 6 |
| $a_{max} = 1.3$ m/s$^2$, $a_{min} = -2$ m/s$^2$, $\tau = 0.2$ $\mu = 32.46$ s, $\sigma = 8.18356$ s | 4 | 36 | 16 | 36 | 21 | 36 |

For example, Figure 4 shows how for $C_e$ = 2, 3 or 4 a higher number of $C_r$ passive reservations must be made, through a fixed circular cluster of $C_r$ = 6, 18 or 36 cells with a radius of $C_e$ cells (including the active one). This introduces huge resource wastages, due to the enormous amount of passive pre-reserved bandwidth over $C_r$ cells, which increases for longer calls or for higher values of $v_{max}$, for fixed values of *CHT* and $v_{max}$, respectively.

If additional information about the mobility directional behavior is available, the issues above can be avoided, with a lower $C_r$ (near or equal to $C_e$). This last opportunity is provided by our proposal, and the obtained values of $C_r$ depend on the adopted reservation thresholds and policies. A generic cell can be well approached by an *n*-edge regular polygon as depicted in Figure 4 (*n* can be considered as an input control parameter and it is commonly equal to 6).

For higher values of *n*, better approximations can be reached. A set $S_{ho}$ (hand-off directions set) of *n* possible movement directions (i.e., hand-off directions) can be obtained: $S_{ho} = \{d_1 \ldots d_n\}$, with $d_j = \theta(2j - 1)/2$ rad., $\theta = 2\pi/n$ rad., $j = 1 \ldots n$ and $|S_{ho}| = n$.

The conditional probability that a mobile host will hand-out to direction $y \in S_{ho}$ after *CST* amount of time (Gaussian distributed), having handed-in the cell from direction $x \in S_{ho}$ can be defined as:

$$M(x,y) = p_{x,y} = pc_{NSIS-MIP}(x,y) = p(\text{out to } y \in S_{ho} \ t = t_0 + CST / \text{in from } x \in S_{ho} \ t = t_0), \tag{13}$$

where $t_0$ is the time at which the user hands-in the considered cell. Once $n$ and $S_{ho}$ have been defined, a square *nxn* HDP matrix $M$ is defined as: $M(x,y) = p_{x,y} = p_{CNSIS-MIP}(x,y)$. To be noticed that $CST \sim N(\mu_{CST}, \sigma^2_{CST})$. The composition of $M$ depends on the adopted mobility model and coverage topology. It has the hand-in and hand-out directions on the rows and columns, respectively; its elements can be obtained through the preliminary campaign of simulations, while acquiring the *CST* distribution. As for the *CST* analysis, the Kolmogorov-Smirnov (KS) normality test was carried out on the $n^2$ elements of $M$. This test is based on the *p*-value concept: it is a measure of how much evidence there is against the null hypothesis; different *p*-values were obtained, showing the goodness of the Gaussian distribution hypothesis (for details, to see [50]). Therefore, from different simulation runs, it resulted that the elements of $M$ follow also a Gaussian distribution, and they can be represented by a mean and a standard deviation $\mu_{p(x,y)}$ and $\sigma_{p(x,y)}$. In this sense, $M(x,y)$ is a couple of values. An example of $M$ is shown in Figure 5.

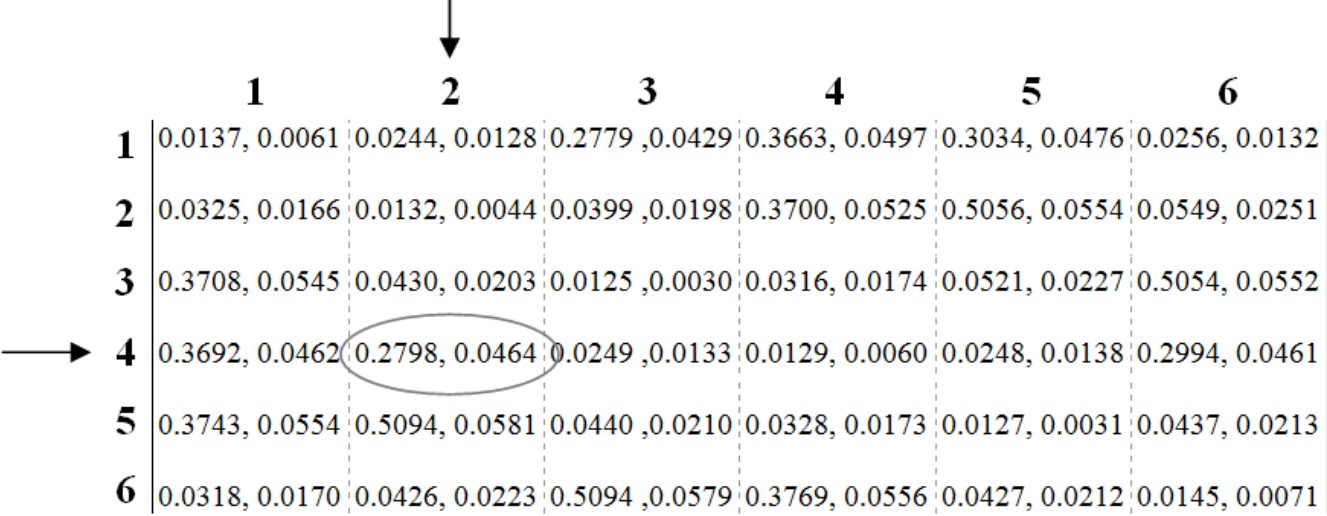

**Figure 5.** The obtained M matrix for the mobility parameters of [42]; for example $M(4,2) = p4,2 = pcNSIS\text{-}MIP(4,2) = N(0.2798, 0.0464)$, where $\mu(4,2) = 0.2798$ and $\sigma(4,2) = 0.0464$.

### 6.1. Static Scheme

The first proposed policy consists of a static prediction scheme that selects the cells for being involved by passive reservations, according to some parameters (maximum speed, average speed, *CST* and HDP). When, for a mobile host $i$, $C_{ei} \geq 2$ (at least one predicted hand-off event), let $j = 1, ..., (C_{ei} - 1)$ be the index associated to the $j$-th hand-off event, where $C_{ei}$ is derived as in [31]; let indicate the number of desired predicted cells for $j$-th hand-off of user $i$ with $C_{ij}$, where $C_{ij} \in \{1, \ldots, n\} \ \forall \ j$.

We propose three reservation schemes for the static case:

(1) Non-decreasing: each user $i$ will reserve on an increasing number of cells for the increasing number of hand-over (i.e., $C_{i1} \leq C_{i2} \leq \ldots \leq C_{iCei-1}$);

(2) Non-increasing: each user $i$ will reserve on a decreasing number of cells for the increasing number of hand-over (i.e., $C_{i1} \geq C_{i2} \geq \ldots \geq C_{iCei-1}$);

(3) Constant-trend: each user $i$ will reserve on the same number of cells for every hand-over ($C_{i1} = C_{i2} = \ldots = C_{iCei-1}$).

When $M$ shows that there are no preferential mobility directions in the analyzed traces, the pre-reservation over only one cell for every hand-over ($C_{ij} = 1 \; \forall j$) may lead to a high Call Dropping Probability (CDP), as illustrated in the next section. This issue can be solved by pre-reserving over multiple hand-out directions: this implies that the values of $C_{ij}$ must be chosen adequately. Once the $C_{ij}$ values are chosen, the proposed scheme uses $M$ to predict the next cell directions $y$ for every $j$-th hand-off event of user $i$. If the current hand-in direction is $x$, then $y = index\{max[M(x)]\}$, where $M(x)$ is the $x$-th row of $M$ and $x, y \in S_{ho}$; this is repeated $C_{ij}$ times for every $j$-th hand-off. For every iteration, previous chosen values are not considered yet when picking up the current maximum. The pseudo-code illustrated in Figure 6 resumes the main steps of the algorithm. It receives $n$ and $C_i = [C_{i1}, ..., C_{ho\text{-}max}]$ as input control parameters: since $C_{ei}$ cannot be known a-priori because it depends on the *CHT* (which is assumed to be exponentially distributed) a maximum number of hand-off events $C_{ho\text{-}max}$ for every call must be considered, under the assumption of $C_{ei} \geq 2$. The algorithm starts the prediction from the active cell, where the call originated.

**DIRECTION AWARE DYNAMIC PREDICTOR**

```
//for each predicted hand-off event of user i
for k=2 to hᵢ {
    //index on the cells of the k-th hand-off event
    l=1;
    //for each cell of the current k-th hand-off event
    while l ≤ vhᵢ[k].size() {
        //analyze the current l-th tuple in the k-th element of vhᵢ
        current_tuple= vhᵢ[k].elementAt[l];
        //the hand-in direction should be determined
        curr_hand_in_dir= From(current_tuple.to);
        //probability of user i of being in current cell after the
        //(k-1)-th hand-off
        pcurr=current_tuple.pcell_id;
        //find the "more suitable" hand-out candidate cells
        //over the n possible hand-out directions
        for p=1 to n {
            //the probability of hand-out on direction p after having
            //handed in on direction curr_hand_in_dir is evaluated
            curr_prob=M(curr_hand_in_dir, p)*pcurr;
            //threshold based comparison
            if curr_prob ≥ δ^f(k) {
                //the current cell can be considered a valid candidate
                id=Cell_id( current_tuple.cell_id, p);
                //the vhᵢ vector must be updated
                create_a_tuple{id, curr_hand_in_dir, p, curr_prob};
                append the tuple in vhᵢ[k+1];
            }
        }
    }
    clean vhᵢ[k+1] from duplicates;
}
create an empty cell identifiers list p_cells;
//extract cell ids from tuples and append them to p_cells
for (int k=1; k<=tᵢ; k++) {
    for (int l=0; l<vhᵢ[k].size(); l++) {
        current_tuple=vhᵢ[k].elementAt(l);
        append current_tuple.cell_id to p_cells;
    }
}
return p_cells.
```

**DIRECTION AWARE STATIC PREDICTOR**

```
take n and Cᵢ as input parameters;
evaluate Cₑᵢ as in [28];
//make the prediction for a max value of Cₕₒ₋ₘₐₓ hand-off events
if Cₑᵢ>Cₕₒ₋ₘₐₓ Cₑᵢ=Cₕₒ₋ₘₐₓ ;
//create the set of predicted handoff directions
p_cells={active_cell};
//for each predicted hand-off event
for j=1 to Cₑᵢ-1 {
    //take from Pcells the set of the pred. cells for the j-th hand-off
    current_Pcells=take_the_j-th_set_of_predicted_cells(p_cells);
    //make a prediction of Cᵢ next cells for all the candidates
    for (every cell in current_Pcells) {
        //if not predicting for the first hand-off event
        if j>1 {
            current_cell=pick_the_next_cell(current_Pcells);
            //evaluate the hand-in direction for the current_cell
            current_x=take_the_current_hand-in_direction(current_cell);
            //remember the elements of M(current_x) already visited
            current_max_set=∅;
            //make the right prediction
            for (int i=0; i<Cᵢⱼ; i++) {
                //evaluate the most probable hand-out direction
                curr_direction_y= index{max[M(current_x)\current_max_set]};
                //evaluate the next cell on current direction y
                p_cells.append(adjacent_cell on direction curr_direction_y);
                //mark the current direction y as already visited
                current_max_set.append(current_direction_y);
            }
        }
        else {
            //evaluate the current moving direction of user i
            dᵢₖ=evaluate_the_current_direction_of_user_i;
            //calculate the most probable Cᵢ₁ cells for first hand-off event on dₖ
            first_set=determine_the_set_of_candidate_cells(Cᵢ₁,dₖ);
            p_cells.append(first_set);
        }
    }
}
return p_cells;
```

**Figure 6.** Pseudo-code of the proposed algorithms.

Given the unavailability of $M$ at the moment of the service request, the first future cell is evaluated by one of the approaches of [51,52]. Assuming that user $i$ will probably follow direction $d_k$ until the first hand-out event, the identifier of the most $C_{i1}$ probable cells that user $i$ will visit following direction $d_{ik}$ from the current position can be discovered

and inserted into the prediction set *Pcells*. From $j = 2$ to $j = C_{ei-1}$, the algorithm creates a temporary set called *current_Pcells* with the predicted cells belonging to *j*-th hand-off event. For each of them, it determines the hand-in direction called *current_x*, then it evaluates the maximum value in the vector $M(current\_x) \backslash current\_max\_set$. The last discovered maximum value is appended into the "*current_max_set*" vector, that is subtracted from $M(current\_x)$ in the next iteration (in this way the new maximum value is always calculated, without considering the previous ones). The algorithm has a time complexity of $O((C_{ei-1}) \cdot n^2)$. Table 1 shows $C_r$ values are obtained through (12) (they belong to the circular reservation policy as depicted in Figure 4), while $C_r(P)$ values are obtained following the approach previously proposed, for different reservation policies $P$ (non-decreasing, non-increasing or constant). However, it can be seen that there is a resource gain if a directional treatment is introduced. For instance and without loss of generality, $C_{ho\text{-}max}$ has been fixed to 3 (under the assumption that the generic call *i* is long enough in order to suffer at least 3 hand-over events). The notation $P(C_1\text{-}C_2\text{-}C_3)$ indicates that the reservation policy $P$ makes passive reservations on $C_1$, $C_2$ and $C_3$ cells for *1-st*, *2-nd* and *3-rd* hand-off, respectively, that is to say the input vector $C$ is $[C_1, C_2, C_3]$, as previously illustrated (*i* is not used for the sake of simplicity).

*6.2. Dynamic Scheme*

In the proposed dynamic scheme, the number of predicted cells for the *j*-th hand-off event are chosen dynamically, through an input control threshold $\delta$. Additionally, in this case $C_{ei}$ is evaluated with the approach of [31]. Let $h_i = C_{ei} - 1$ be the number of hand-over events of user *i*. Let $v_{hi}$ be an array, whose elements $v_{hi}[k]$ $(k = 1 \dots h_i)$ indicate the information about the *k*-th future hand-off of user *i*. Each entry $v_{hi}[k]$ can be a pointer to, for example, a list of tuples {cell_id, from, to, $p_{cell\_id}$} for the *k*-th hand-off event, with:

(a) cell_id is a cell identifier;
(b) from, to $\in S_{ho}$ are the hand-in and hand-out directions for the *cell_id*;
(c) $p_{cell\_id}$ is the probability that user will be covered by *cell_id* after the *k*-th hand-off.

The algorithm predicts to directions for each tuple, starting from *cell_ids*, from directions and $p_{cell\_id}$ values. Let $\delta$ be an input threshold: if the knowledge of the first hand-off cell is approached, then the threshold-based predictor illustrated in Figure 6 can be activated, for obtaining the complete set of cells that will be visited by the NSIS-MIP *i*-th user.

With one of the approaches of [51,52], the current direction $d_j \in S_{ho}$ for user *i* is discovered and the term *first_id = first_Cell_id(current_id, $d_j$)* can be obtained, by the function *first_Cell_id*, which evaluates the identifier of the cell that user *i* will visit. Th e error introduced by this approach is negligible (around 3–4%). At this point, a tuple {*first_id, _, $d_j$, 1*} can be created and appended in $v_{hi}[1]$ (the 'from 'direction cannot be discovered because user *i* has started its flow in the current *first_id* cell and $p_{first\_id} = 1$, because the probability of handing-out from *first_id* cell during the first hand-off is 1).

For sake of simplicity, let us assume for now that the elements of *M* are constant values. As shown earlier, each tuple in $v_{hi}[k]$ contains the hand-in direction, the cell identifier and the probability of user *i* of being in the cell after the $(k-1)$-th hand-off. Now, a threshold-based comparison is used to decide what are the cells that user *i* will visit with higher probability, when handing-out the cell of the *l*-th tuple of $v_{hi}[k]$, $l = 1 \dots v_{hi}[k]$. size(), with a well-known hand-in direction. The hand-in direction curr_hand_in_dir belongs to $S_{ho}$ and it specifies a unique row of *M*.

The proposed idea calculates the probability of handing-out from the current cell on direction $p$ after having handed-in from direction *curr_hand_in_dir* when the probability of being in the previous cell before the current hand-off is $p_{curr}$. If the obtained value is higher than $\delta^{f(k)}$, then the cell that is adjacent to the current one on direction $p$ must be considered as a possible future cell and a tuple {*adjacent_p_cell, from, p, curr_prob*} is appended in $v_{hi}[k + 1]$.

We provided to add a power operation to $\delta$ in order to account for the prediction error increasing for higher values of *k*. The function "cell_id Cell_id (cell_id current_id, direction

to)" returns the identifier of the cell adjacent to *current_id* cell on *to* direction. The function "direction from (direction to)" translates the hand-out direction *to* of the previous cell in the hand-in direction of the next cell. When repeating all the steps $h_{i-1}$ times, a cleaning routine must be executed after finishing appending elements in $v_{hi}[k]$ position, because of possible duplications of cell identifiers.

The same results can be obtained if the "append" function avoids duplicates. This algorithm has a time complexity of $O((C_{ei}-1)\cdot n^2)$. The prediction result is the set of cell identifiers of the tuples for each $v_{hi}$ list. Remembering that the hypothesis of $M$ composed of constant values is not suitable, because of $M(x,y)$ consists of a couple of values, we have $M(x,y) = N(\mu_{x,y}, \sigma_{x,y})$.

Four different expressions have been considered for $f(k)$: (a) $f(k) = 1$; (b) $f(k) = \alpha k$; (c) $f(k) = \alpha/k$ and (d) $f(k) = (\alpha k)^{-1}$, with $\alpha > 0$, in order to appreciate the different behavior of the algorithm by varying the $\delta^{f(k)}$ structure and how $\delta$ is weighted for consecutive values of $k$.

## 7. Performance Evaluation

Many simulations were carried out in order to evaluate the performance of the proposed idea in terms of average prediction error, number of involved cells and system utilization.

### 7.1. General Simulation Setup and Parameters

Our network consists of 65 coverage cells, with a coverage radius of about 250 m. The considered geographical region (south of Italy) is a 2.44 km$^2$ area (Figure 7), and users move according to C4R and CityMob; the APs are wired connected, by a switching subnet, to the net-sender. The performances of the rate adaptation and CAC schemes for NSIS-MIP and NSIS-MDP users are also taken into account, in order to evaluate their conformance to QoS parameters (outage probability, minimum received utility values and a high system utilization). All the results have been obtained by following the theory of the confidence intervals as illustrated in [50,53]: the number of runs has been set to $N_{sim}$ = 15, with a simulation time of $T_{sim}$ = 400 s. In this way the committed error while considering simulation results has been limited to a maximum value of 5% (confidence of 95%). An exponentially distributed CHT with mean $\lambda$ = 180 s was considered. Traffic load was fixed to 15 requests/s because it guarantees a good level of system saturation. When a session starts, packets are generated as in [54], based on Poisson and Pareto distributions using a file size distribution heavy-tailed with Pareto parameter fixed to 1.85. Each AP has a total bandwidth of 11 Mbps and users can receive discrete resource levels, from 512 Kbps up to 896 Kpbs (with a gap level of 128 Kbps). The considered utility functions are the same of those illustrated in Section 5. It must be outlined that the current AP dynamically choses the right value of $B$. In our simulations, traffic load is composed of NSIS-MIP and NSIS-MDP flows in variable percentage. The bandwidth is managed by the policies illustrated in Section 5 and the $p_{outage}$ value is set to 0.05.

There are no rules about choosing the value of $n$: simulations results showed that $n$ = 6 is a good trade-off between accuracy and computational complexity of the proposed algorithm; higher values of n make better the approximation of the wireless cell coverage area but make worse the spent computational time. Simulation results of the dynamic-scheme are compared with those of the static-scheme and the obtained enhancements are shown. First of all, the HDP matrix $M$ was filled up, so a certain number of monitor simulations were launched; in particular 1000 monitor runs were executed with a single duration of $T_{sim}$ = 400 s, in order to obtain a smoother distribution curve. So, after a statistical analysis of the obtained values with MATLAB tool, the matrix M has been generated such as shown in Figure 5. The static algorithm was tested with the following input parameters:

(1)  constant trend: $C_{i1} = C_{i2} = C_{i3} = 1$;
(2)  increasing trend: $C_{i1} = 1$, $C_{i2} = 2$, $C_{i3} = 3$;

(3)    decreasing trend: $C_{i1} = 3$, $C_{i2} = 2$, $C_{i3} = 1$ with $C_{ij} = 1 \; \forall j \in \{4 \ldots C_{ei-1}\}$;

The input values of $C_{ij}$ are different from those of Section 6 because attention was focused only on the first, the second and the third hand-off events.

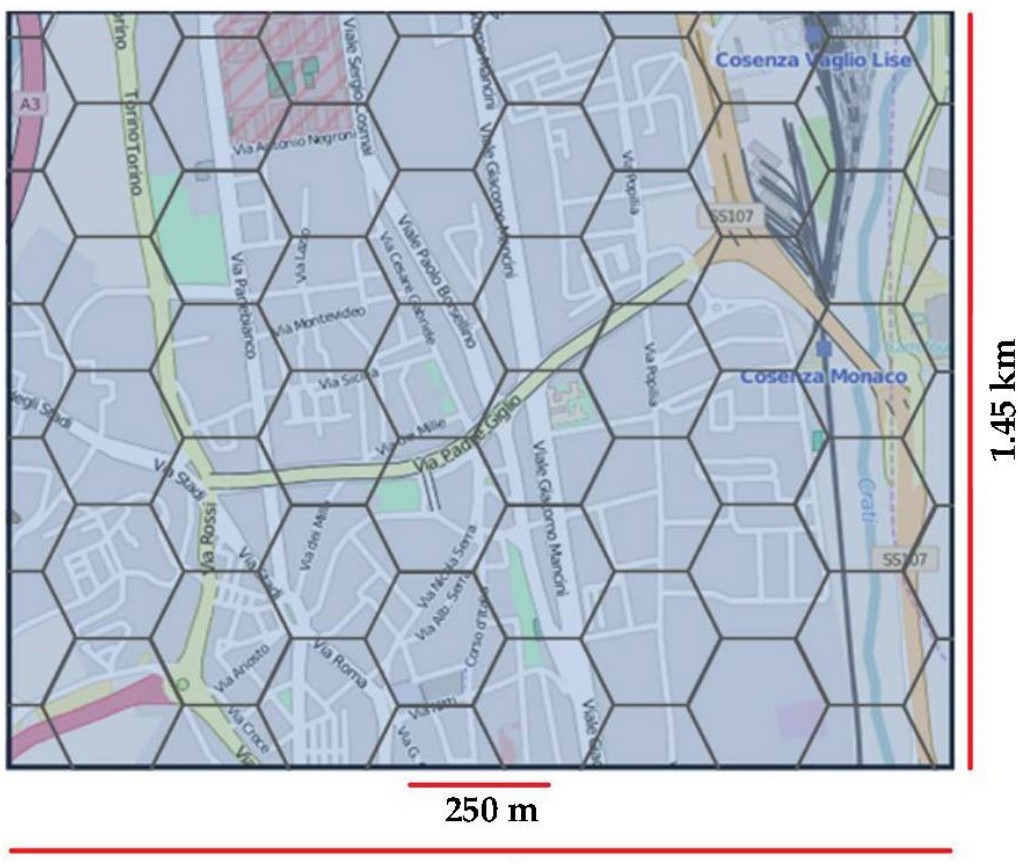

**Figure 7.** The simulated map.

In this section, only the results for case (b) of Section 6, with $\Delta = \delta^{\alpha k}$ are considered, because the employing of the $\alpha k$ exponent into the dynamic algorithm leads to better results. After a deep analysis of the possible values of $\alpha$ the value $\alpha = 1$ was chosen, because it guarantees the optimal performances for the chosen exponent function. As for the monitor simulations, the duration was fixed to $T_{sim} = 400$ s for each run. Different campaigns were carried out, also varying the amount of NSIS-MIP and NSIS-MDP traffic percentages; in the following, if 60% is the NSIS-MIP percentage then, obviously, 40% is the percentage of NSIS-MDP traffic.

### 7.2. Main Reachable Results

Figure 8 illustrates how the utility is perceived by users by varying the NSIS-MIP traffic percentages and some prediction input parameters. As illustrated later, it must be outlined that the wireless channel evolution does not depend on the adopted prediction policy, so it has a similar evolution behavior for both static and dynamic case. In fact, in both cases (static and dynamic), if the percentage of NSIS-MIP flows increases the perceived utility is higher, due to the lower number of admitted NSIS-MDP flows, which makes the bandwidth availability higher. Additionally, in this case the static prediction policy offers better performances, while low values of delta are preferred if a dynamic policy is pursued.

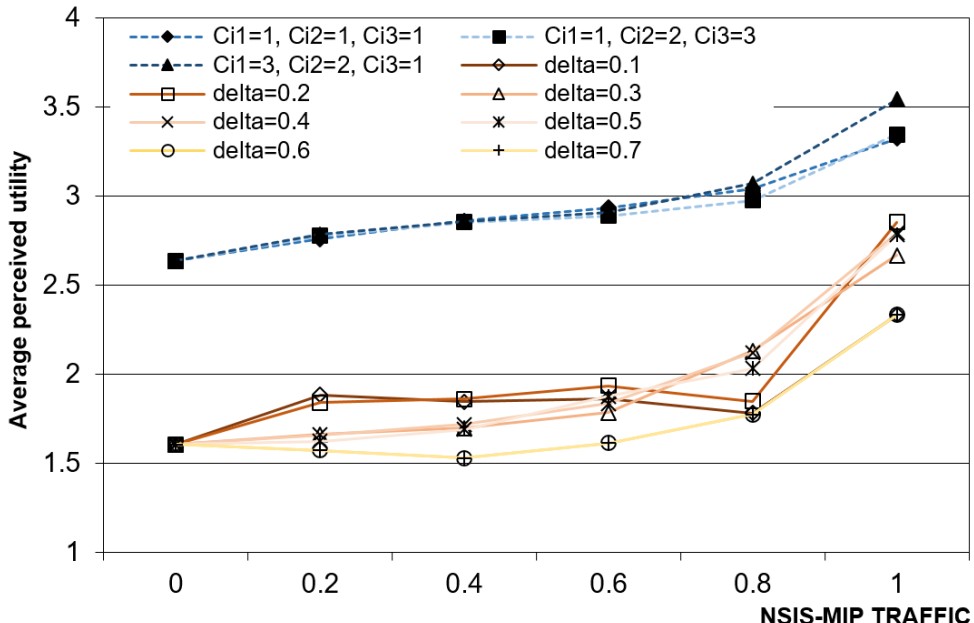

**Figure 8.** Average perceived utility vs NSIS-MIP traffic percentage.

System utilization for increasing values of NSIS-MIP requests is shown in Figure 9. In both static and dynamic cases, the trend is decreasing because there is a higher presence of passive reservations, and a consequent reduction of NSIS-MDP admitted flows. For NSIS-MIP service requests equal to 0, the utilization reach its maximum value (about 92–93%), because the system will not provide passive reservations. On the other side, the system becomes under-utilized because only NSIS-MIP users make service requests. The maximum gap between static and dynamic schemes is observed for an NSIS-MIP percentage of 60% and it is about 10–12%. The performance of the static scheme are slightly better, except for high percentages of NSIS-MIP flows, because high values of $\delta$ lead to a system utilization near to 46%.

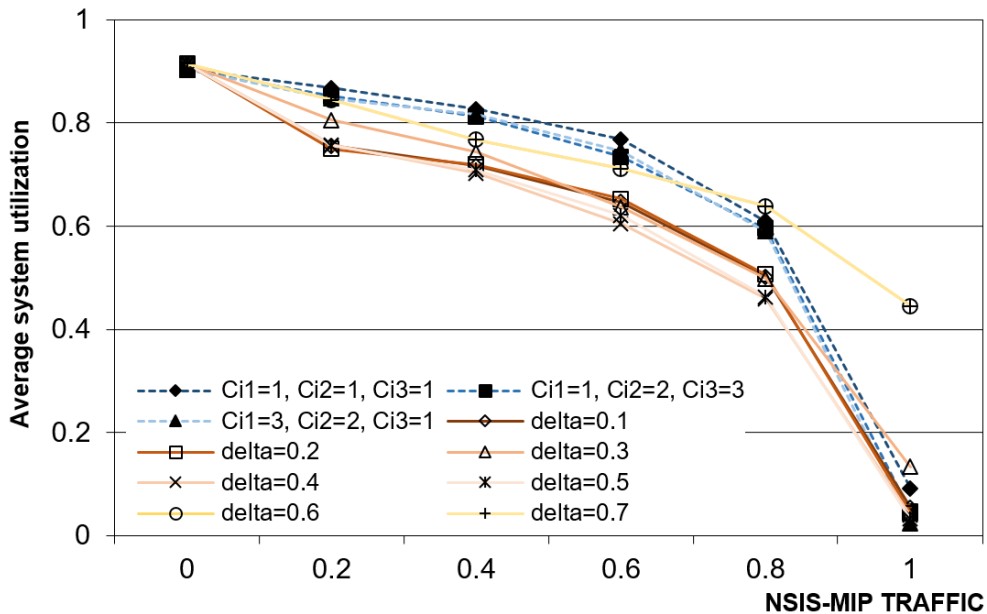

**Figure 9.** Average system utilization vs NSIS-MIP traffic percentage.

Figures 10 and 11 depict the average trend of the number of NSIS-MIP and NSIS-MDP admitted flows, respectively; both in the static case and in the dynamic one there is an obvious increasing behavior of NSIS-MIP admissions for a higher number of NSIS-MIP requests, while the NSIS-MDP ones drastically decrease. For the static prediction case the number of admitted NSIS-MIP flows is often lower than 200 while, in the dynamic case, for $\delta = 0.3$, $\delta = 0.6$ and $\delta = 0.7$ the number of admitted NSIS-MIP flows is not comparable, because it increases to about 780. Both in the static or dynamic cases the NSIS-MDP admission is not affected by the chosen policy and the chosen input parameters. The dynamic algorithm ensures higher admission chances for NSIS-MIP flows, in spite of NSIS-MDP flows that find a lower amount of available bandwidth if more NSIS-MIP requests are accepted.

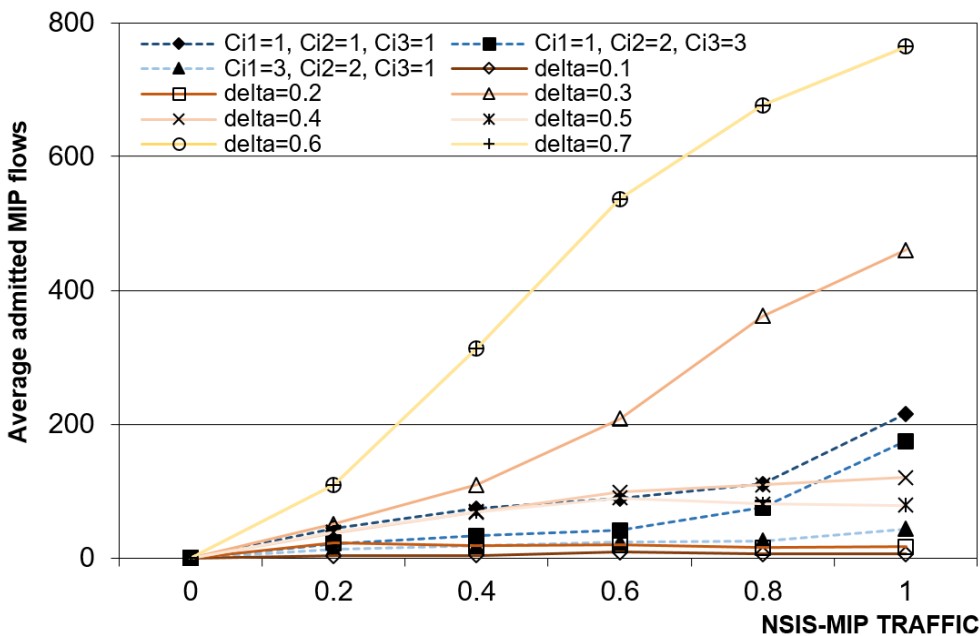

**Figure 10.** Average number of NSIS-MIP admitted flows vs NSIS-MIP traffic percentage.

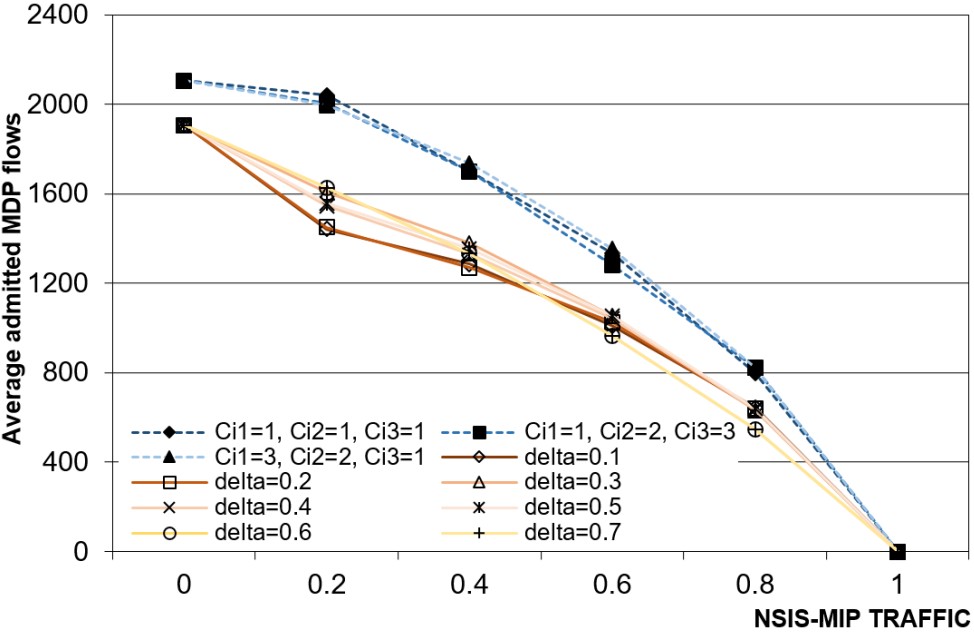

**Figure 11.** Average number of NSIS-MDP admitted flows vs NSIS-MIP traffic percentage.

Figure 12 illustrates the average number of dropped NSIS-MIP flows: these curves show how the CAC module and the bandwidth reallocation algorithm guarantee a very low dropping probability to NSIS-MIP flows. In the dynamic prediction case, the number of NSIS-MIP dropped flows is higher than the static one, but if it is compared with the average number of admitted NSIS-MIP flows, the percentage of NSIS-MIP dropped flows maintains at a very low and acceptable level, as well as in the static case. The minimum and maximum NSIS-MIP dropping observed percentages for the static case are 0.0285% and 0.3%, while for the dynamic prediction case they are 0.31% and 0.87%. It can be concluded that both prediction schemes with the CAC and bandwidth management of Section 5 have good performances in terms of percentage of NSIS-MIP dropped flows, which is maintained below 1%.

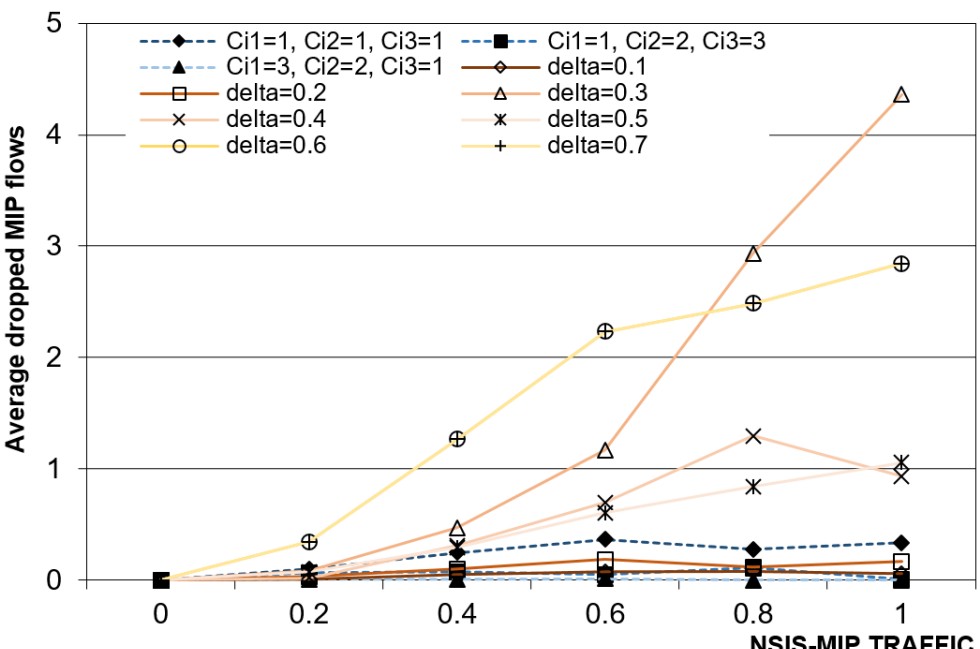

**Figure 12.** Average number of NSIS-MIP dropped flows vs NSIS-MIP traffic percentage.

For NSIS-MDP users (not shown) the proposed bandwidth management algorithm gives only an outage intra-cell guarantee, so they may not find the needed bandwidth after a hand-off event; in addition if an NSIS-MIP user need some bandwidth to mitigate its outage condition, the algorithm drops an NSIS-MDP flow if no bandwidth is available.

In Figure 13 the average error committed in the prediction of possible next cells for the third hand-off event of NSIS-MIP users in the static and dynamic cases can be seen. If $N_{NSIS\text{-}MIP}(ho)$ is the overall number of NSIS-MIP users who have made at least ho hand-overs from the first cell and $n_{NSIS\text{-}MIP}(ho)$ is the overall number of NSIS-MIP users who did not find a passive reservation after the ho-th hand-off event, then the prediction error on ho-th hand-off event is $e(ho) = n_{NSIS\text{-}MIP}(ho)/N_{NSIS\text{-}MIP}(ho)$. In addition, for a fixed prediction scheme with a fixed set of input parameters the trend is almost constant if the NSIS-MIP traffic percentage is varied; obviously, the prediction algorithm is not affected by the number of admitted flows. Some input combinations must be excluded, because they do not lead to any acceptable result. The best results were obtained for the dynamic case with $\delta = 0.5$ and they are not comparable with those obtained with any other input values of the dynamic scheme or with the static one, which has good performances for the input sequence $C_{i1} = 1$, $C_{i2} = 2$, $C_{i3} = 3$. The minimum and maximum error for $\delta = 0.5$ are 10% and 12.57%.

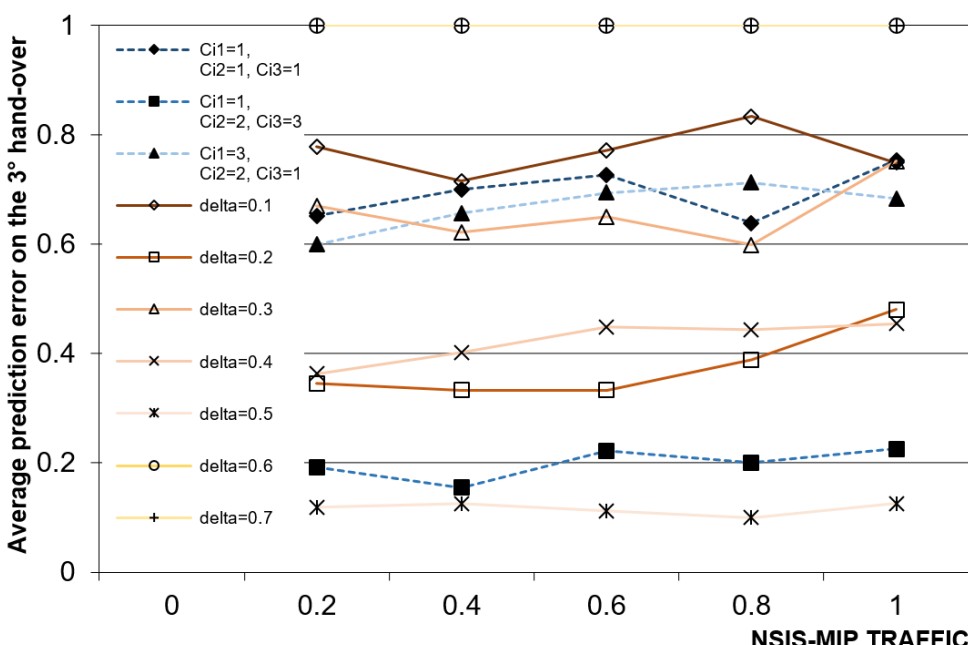

**Figure 13.** Average prediction error on 3° hand-off vs. NSIS-MIP traffic percentage.

Table 2 shows the average percentage of NSIS-MDP users that suffered an outage event during their active sessions (NSIS-MIP analysis is not shown because the outage percentage resulted less than 0.1% for every combination of scheme/parameters): in our simulations the outage threshold was fixed at $p_{outage}$ = 0.05. The implemented CAC and bandwidth allocation algorithm ensure the threshold constraint to be respected: in fact, the maximum observed value of outage percentage for NSIS-MDP users is 0.0502%. In particular, as discussed earlier, NSIS-MIP users are privileged when compared to NSIS-MDP ones, because of the guaranteed service continuity and the passive reservation policy.

**Table 2.** Average NSIS-MDP outage percentage vs NSIS-MIP traffic percentage.

| | STATIC | | | DYNAMIC | | | | | | |
|---|---|---|---|---|---|---|---|---|---|---|
| **NSIS-MIP** | **111** | **123** | **321** | **0.1** | **0.2** | **0.3** | **0.4** | **0.5** | **0.6** | **0.7** |
| 0 | 0.01213 | 0.01275 | 0.01542 | 0.01547 | 0.01551 | 0.01558 | 0.01571 | 0.01583 | 0.0163 | 0.018 |
| 0.2 | 0.01433 | 0.01455 | 0.01143 | 0.01278 | 0.01358 | 0.01945 | 0.01466 | 0.01448 | 0.01479 | 0.01483 |
| 0.4 | 0.01913 | 0.01874 | 0.01893 | 0.01932 | 0.01945 | 0.02013 | 0.01957 | 0.01961 | 0.01968 | 0.01973 |
| 0.6 | 0.0214 | 0.01973 | 0.01995 | 0.0214 | 0.01903 | 0.02988 | 0.02531 | 0.0174 | 0.02243 | 0.02673 |
| 0.8 | 0.0403 | 0.0413 | 0.042 | 0.0416 | 0.0435 | 0.0448 | 0.04532 | 0.04723 | 0.04831 | 0.04981 |
| 1 | | | | | | | | | | |

In addition the bandwidth allocation algorithm privileges NSIS-MIP users when choosing a benefactor who has to give up a portion of his allocated resources (which is why the percentage increases for higher NSIS-MIP traffic). In terms of outage percentage of NSIS-MDP flows (also for NSIS-MIP ones) no large differences can be observed between static and dynamic schemes. The increasing trend of the values in Table 2 does not depend on the adopted prediction policy, but is due to the higher presence of NSIS-MIP flows, which have a certain allocation priority. From the figures above it can be concluded that both static and dynamic prediction schemes are able to guarantee a good level of QoS for different input parameters; in particular, the static scheme is able to ensure a slightly higher level of utility (Figure 8) with an acceptable system utilization (Figure 9).

The percentage of dropped NSIS-MIP flows is also maintained under the value of 1%. The best obtained input sequence for the static scheme is $C_{i1}$ = 1, $C_{i2}$ = 2, $C_{i3}$ = 3; this can be explained by considering the increasing in prediction error for higher hand-off events

due to the intrinsic error that was committed in the generic previous step. Pre-reserving resources on a higher number of cells for the next hand-off event can balance previous prediction errors. The dynamic scheme offers slightly lower performances in terms of amount of assigned bandwidth and perceived utility, but it outperforms the static one in terms of prediction error. For the first hand-off only the static sequence $C_{i1} = 3$, $C_{i2} = 2$, $C_{i3} = 1$ leads to a negligible value of $e(1)$, because reserving on $C_{i1} = 3$ cells reduces the probability of error near to zero. However, the dynamic threshold-based algorithm performs better in the "long-range" prediction: the maximum value of $e(3)$ is 12.57% for $\delta = 0.5$.

### 7.3. Performance Comparison

In this sub-section, our proposed idea is compared with other two ones: the User Mobile Profile (UMP, as already discussed for [12]) and the Active Lempel-Ziv (ALeZi) algorithm [55], with the aim of showing the advantages introduced by a passive reservation approach and if the proposed idea could be better than other ones. The UMP algorithm evaluates the probabilities of each cell that the user can visit in the future by using three main data structures: the Trace Record Matrix (TRM), the Path Database (PD) and the Historical Path Database (DH). It is also able to perform a multi-step prediction, in the sense that it can predict the probability of visiting cells at the $k$-th hand-over event (with $k = 1, \ldots, n$ where $n$ is the number of hand-overs that the mobile host will perform). Obviously, given the visiting probabilities determined by UMP, if more cells are considered in the prediction, then a better approximation will be obtained, at the price of a bigger resource wastage (in the figures, UMP-2 and UMP-3 indicate the UMP with 2 or 3 passive reservations for each next cell). The AleZi scheme, instead, represents a single-step predictor (for each prediction step, only the first next cell is predicted), based on the Lempel-Ziv algorithm. Each cell of the system will be represented by a symbol, so the sequence of the previously visited cells will be represented by a string, used to predict the most probable next symbol (i.e., the next cell), by the Prediction by Partial Matching (PPM) approach. For the static and dynamic schemes the best cases have been considered (that is the sequence $C_{i1} = 1$, $C_{i2} = 2$, $C_{i3} = 3$ and $\delta = 0.5$, respectively). We provided to consider the same topology of the previous simulations.

Figure 14 shows the trend of the average system utilization offered by the four different algorithms, in function of the number of requests per second (only for MIP traffic).

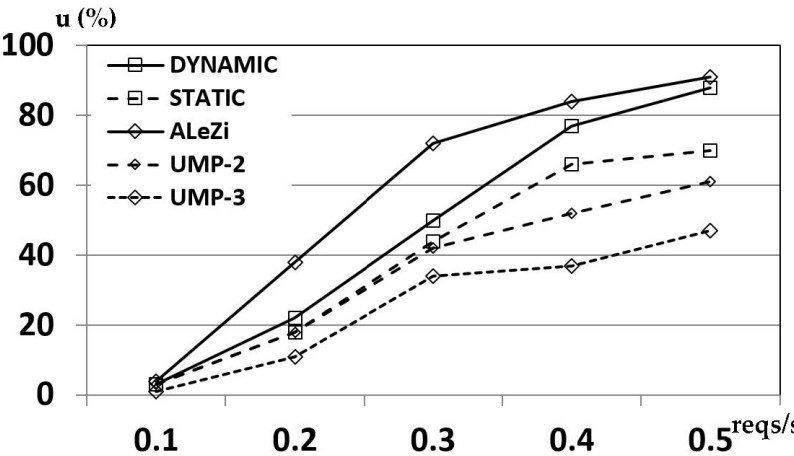

**Figure 14.** Average system utilization for the considered predictive algorithms.

It can be seen that the ALeZi scheme outperforms the other three algorithms, because it is able to reserve passive bandwidth only on the single predicted next cell, so there is a very limited presence of passive reservations into the system, leading to a balanced level of active bandwidth (perceived by mobile hosts while served by a coverage cell). Clearly, the

absence of a multi-step approach, makes the ALeZi weak in terms of service continuity, given that the mobile hosts could reach a cell in which there are no passive reservations for it. Our dynamic proposal, offers a good trade-off in terms of system utilization: it cannot perform as good as ALeZi, but the possibility to dynamically choose the number of cells over which the passive reservations should be made gives the opportunity to be near ALeZi for higher MIP requests. As regards UMP-2 and UMP-3, their utilization is very low when 2 or 3 cells are considered to be influenced by passive reservation.

Figure 15, instead, shows the average prediction error committed during the whole CHT of mobile hosts. It is evident that a single-step predictor (such as ALeZi) is not able to guarantee a good performance for this metric, given that only the next cell is predicted for each hand-over event, so it fails to capture the dynamics of mobile users (no memory effect).

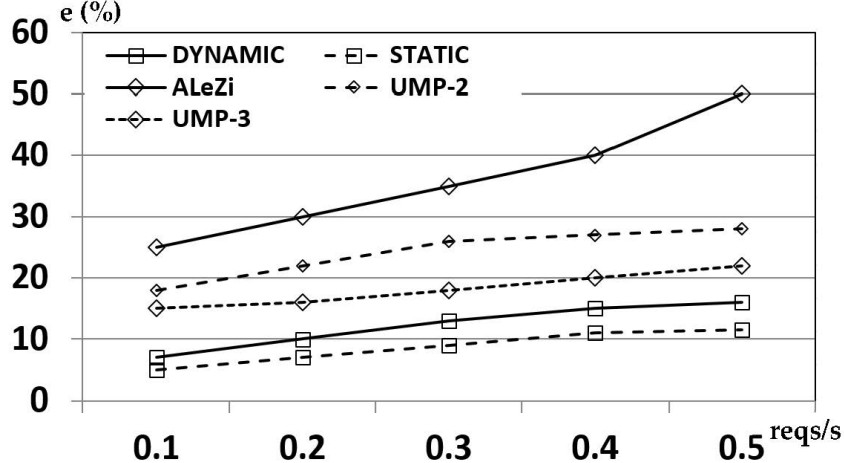

**Figure 15.** Average prediction error related to the considered predictive algorithms.

Additionally, in this case, our dynamic scheme outperforms the UMP, with a maximum error of 16.8% for higher service request rate.

In Figure 16, the average Call Dropping Probability (CDP, the probability of a forced termination of a call after an hand-over event) is clearly unacceptable for the ALeZi algorithm: given that it predicts only one next cell (if enough bandwidth is available), the probability of finding a saturated/congested coverage cell is much higher. For the other schemes, no big differences are visible (their CDP ranges from 0.081, almost constant for the static predictor, to 0.123 for the UMP-2 approach, so for the CDP metric they are quite similar.

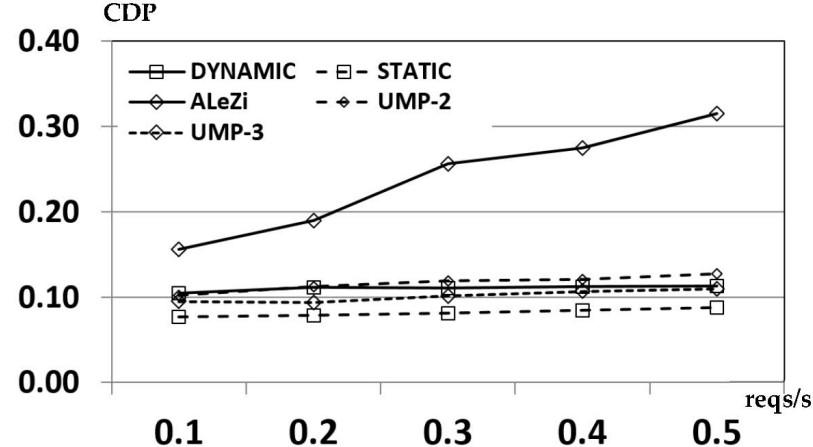

**Figure 16.** Average Call Dropping Probability (CDP) values for the considered predictive algorithms.

## 8. Conclusions

This work addresses some important problems that play a key role in wireless environments with cellular coverage. There has been a lot of research and development in wireless networking, and the main aim has always been the mitigation of wireless effects, improving the quality of the offered services. This work makes a contribution in this field by taking into account wireless propagation phenomena and mobility management.

A utility-oriented algorithm has been proposed for the CAC and bandwidth allocation; it considers wireless link conditions, giving more effectiveness to the obtained results. For the mobility management purpose, two prediction schemes have been proposed: the first is based on a static scheme, while the second has a different behavior, because it applies a dynamic technique that chooses the predicted cells. The NSIS with some extensions for the dynamic bandwidth re-assignment has been used for the flows management in a 2D wireless system, while user mobility has been modeled through CityMob and C4R. Two service classes have been considered: NSIS-MIP services request more guarantees than NSIS-MDP ones, because they have to ensure no degradations or droppings during hand-off events. Many simulations campaigns were carried out in order to verify the correctness and efficiency of the proposed idea. The obtained results have shown that the pre-reservation phase is suitable in wireless environments if a certain level of QoS and service continuity must be ensured (the NSIS-MDP dropping percentage is too much higher if compared with the one of the NSIS-MIP traffic, which is lower negligible).

So, the pre-reservation phase is mandatory and two prediction schemes (static and dynamic) have been proposed, in order to realize the "passive reservations" policy. Both prediction algorithms have shown satisfactory results, but the dynamic one led to a more accurate "long-term" prediction for some specific values of input parameters. In particular, in terms of system utilization, our static/dynamic proposals cannot outperform a single-step predictor, given its very low number of passive reservations (the maximum gap is about 26.3% between static and ALeZi), but in terms of prediction error it ranges from 4.5% to 15.7%, completely outperforming the other considered algorithms (which range from 15.1% to 50.5%). Additionally, in terms of CDP, the static predictor maintains below 0.1, while the dynamic one reaches some results comparable with the other multi-step scheme, while ALeZi is completely outperformed. By the obtained results, we confirmed that the proposed predictive schemes (the dynamic in particular), are able to adapt to specific mobile scenario by setting a proper threshold and the input parameters, guaranteeing an optimized management of the available resources.

As future works, we will consider naturalistic GIS applications, able to fix the errors in mobility samples collection and to obtain more realistic traces. Moreover, it would be interesting taking into consideration studies on human mobility traces for constructing mobility models and evaluating mobile opportunistic communication throughout the use of social networks such as [56].

**Author Contributions:** Conceptualization, P.F. and M.T.; methodology, M.T.; software, P.F. and M.T.; validation, P.F. and M.T.; formal analysis, M.T.; investigation, P.F.; writing—original draft preparation, M.T.; writing—review and editing, P.F.; visualization, M.T.; supervision, P.F. Both authors have read and agreed to the published version of the manuscript.

**Funding:** This research received no external funding.

**Institutional Review Board Statement:** Not applicable.

**Informed Consent Statement:** Not applicable.

**Data Availability Statement:** Not applicable.

**Acknowledgments:** We kindly thank Katia Giovinazzo for her effort in modeling utility functions related to traffic data.

**Conflicts of Interest:** The authors declare no conflict of interest.

## Abbreviations

The following abbreviations are used in this manuscript:

| | |
|---|---|
| ALeZi | Active Lempel-Ziv |
| AP | Access Point |
| BE | Best Effort |
| BER | Bit Error Rate |
| BRS | Bandwidth Reallocation Scheme |
| BS | Base Station |
| C4R | Citymob4Roadmaps |
| CAC | Call Admission Control |
| CAT | Call Arrival Time |
| CDP | Call Dropping Probability |
| CHT | Call Holding Time |
| CST | Cell Stay Time |
| DH | Historical Path Database |
| DRSVP | Dynamic resource ReServation Protocol |
| FSMC | Finite State Markov Chain |
| GIS | Geographic Information System |
| HDP | Hand-off Directions Probabilities |
| HMM | Hidden Markov Models |
| ISPN | Integrated Services Packet Network |
| KS | Kolmogorov Smirnov |
| MDP | Mobility Dependent Predictive |
| MIG | Mobility Independent Guaranteed |
| MIP | Mobility Independent Predictive |
| MRSVP | Mobile resource ReSerVation Protocol |
| NSIS | Next Steps in Signaling |
| NSLP | NSIS Signaling Layer Protocol |
| NTLP | NSIS Transport Layer Protocol |
| ODbL | Open Database License |
| OSMF | Open StreetMap Foundation |
| PD | Path Database |
| PoI | Point of Interest |
| PPM | Prediction Partial Matching |
| QoS | Quality of Service |
| RMM | Random Way Point Mobility Model |
| RSVP | ReSerVation Protocol |
| RTV | Real Time Video |
| SS | Switching Subnet |
| TRM | Trace Record Matrix |
| UMP | User Mobile Profile |

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
