# Peer review of "Advanced Resources Reservation in Mobile Cellular Networks: Static vs. Dynamic Approaches under Vehicular Mobility Model"

_telecom, doi:10.3390/telecom2040020_

Round 1

Reviewer 1 Report

In this paper, the authors proposed a prediction technique for the management of mobile services in Wireless Cellular Networks. The proposed method is based both on the analysis of Cell Stay Time and on the direction probabilities of hand-in and hand-out events of mobile nodes from wireless cells. Two types of services are considered in the proposed model, namely the NSIS-MIP and NSIS-MDP. They analyzed user mobility by considering a realistic 2D mobility model. Predictive reservation and admission control schemes are also integrated into the proposed method. The authors also conducted a lot of experiments to experiments to validate their models.

Although the submitted manuscript is well-organized and detailed presented, my major concerns of this paper include: the redaction of this paper, the novelty and contribution of the proposed methods, and the technical depth of the models. My detailed comments are as follows.

Strength of the paper first:

  1. The structure of this paper is clear. To make a better understanding of their design of the architecture, the authors provide figures and a flow diagram.

  1. The descriptions for the design of the system are very detailed.

  1. The authors did plenty of experiments from different aspects to demonstrate the advantages of their model. For example, they used five different metrics to evaluate the performance of their proposed method and conducted sufficient comparative experiments.

Some problems:

  1. The redaction of the paper shall be revised. For instance, the paper is not well segmented, for there is only one paragraph in each section, which makes it difficult for readers to get straight to the main points. Additionally, it would be better if the contributions of the paper are specified in the introduction section.
  2. For the numerical results for evaluation, the authors only provided simulation results of their proposed scheme in different scenarios, which is not convincing enough to prove the significance of the contribution of the paper. In order to have a better illustration of the enhancement of the proposed method, the authors shall conduct more comparison experiments with other methods.
  3. For the reference section, the authors have provided 47 reference papers in total, among which none were published after 2016. It is better to cite some recent works related to the paper, e.g., “Improving distributed anti-flocking algorithm for dynamic coverage of mobile wireless networks with obstacle avoidance,” in Knowledge-Based Systems, “Cooperative Sweep Coverage Problem with Mobile Sensors,” in IEEE Transactions on Mobile Computing (TMC), and “FIS-RGSO: Dynamic Fuzzy Inference System Based Reverse Glowworm Swarm Optimization of energy and coverage in green mobile wireless sensor networks,” in Computer Communications.
  4. The layout of the paper should be revised. For instance, in page 8, there is a large blank space. And in page 15, the caption of Fig.7 should be just under the figure instead of being on the left side of the figure.
  5. In the results section, all lines in the figures are black. In Fig.14, readers can hardly distinguish the performance of each simulation. It would be clearer if the lines representing different simulations are of different colors. Or the authors may add a table to present the results in different cases.
  6. The formatting of the pseudo code should be improved.

Author Response

In this paper, the authors proposed a prediction technique for the management of mobile services in Wireless Cellular Networks. The proposed method is based both on the analysis of Cell Stay Time and on the direction probabilities of hand-in and hand-out events of mobile nodes from wireless cells. Two types of services are considered in the proposed model, namely the NSIS-MIP and NSIS-MDP. They analyzed user mobility by considering a realistic 2D mobility model. Predictive reservation and admission control schemes are also integrated into the proposed method. The authors also conducted a lot of experiments to experiments to validate their models. Although the submitted manuscript is well-organized and detailed presented, my major concerns of this paper include: the redaction of this paper, the novelty and contribution of the proposed methods, and the technical depth of the models. My detailed comments are as follows. Strength of the paper first:

The structure of this paper is clear. To make a better understanding of their design of the architecture, the authors provide figures and a flow diagram.

The descriptions for the design of the system are very detailed.

The authors did plenty of experiments from different aspects to demonstrate the advantages of their model. For example, they used five different metrics to evaluate the performance of their proposed method and conducted sufficient comparative experiments.

Some problems:

  • The redaction of the paper shall be revised. For instance, the paper is not well segmented, for there is only one paragraph in each section, which makes it difficult for readers to get straight to the main points. Additionally, it would be better if the contributions of the paper are specified in the introduction section.

We thank the reviewer for his/her observation. We provided to add some sub-sections (in sections 2, 3 and 7) and the main contributions of the paper have been moved at the end of the introduction section.

  • For the numerical results for evaluation, the authors only provided simulation results of their proposed scheme in different scenarios, which is not convincing enough to prove the significance of the contribution of the paper. In order to have a better illustration of the enhancement of the proposed method, the authors shall conduct more comparison experiments with other methods.

We are grateful to the reviewer. We provided to consider other predictive schemes: to this aim, we provided to consider the proposals of [12] and [55] (new reference), in which a multi-step and single-step predictive approaches have been presented. Section 7.3 has been entirely added into the paper, in order to show the goodness of our proposal, when compared with the considered schemes (the considered metrics are system utilization, prediction error and call dropping probability).  

  • For the reference section, the authors have provided 47 reference papers in total, among which none were published after 2016. It is better to cite some recent works related to the paper, e.g., “Improving distributed anti-flocking algorithm for dynamic coverage of mobile wireless networks with obstacle avoidance,” in Knowledge-Based Systems, “Cooperative Sweep Coverage Problem with Mobile Sensors,” in IEEE Transactions on Mobile Computing (TMC), and “FIS-RGSO: Dynamic Fuzzy Inference System Based Reverse Glowworm Swarm Optimization of energy and coverage in green mobile wireless sensor networks,” in Computer Communications.

We thank the reviewer for his/her observation: the three papers have been properly considered and included in the related work section accordingly. 

  • The layout of the paper should be revised. For instance, in page 8, there is a large blank space. And in page 15, the caption of Fig.7 should be just under the figure instead of being on the left side of the figure.

We are very grateful to the reviewer, all the indicated layout issues have been fixed.

  • In the results section, all lines in the figures are black. In Fig.14, readers can hardly distinguish the performance of each simulation. It would be clearer if the lines representing different simulations are of different colors. Or the authors may add a table to present the results in different cases.

We say thanks to the reviewer. We provided to delete Fig. 14 and to add Table 2 instead, which gives a clearer view of the obtained values.

  • The formatting of the pseudo code should be improved.

We say thanks to the reviewer. We provided to change the formatting of the static and dynamic algorithms pseudo code: the indentation has been added, as well as the indications of the start and end of the cycles.

Reviewer 2 Report

 Advanced Resources Reservation in Mobile Cellular Networks: Static vs Dynamic Approaches Under Vehicular Mobility Model

In this paper, the authors analyze the management of mobile services in Wireless Cellular Networks. In particular, they consider two classes of service: NSIS-Mobility Independent Predictive (NSIS-MIP) and NSIS-Mobility Dependent Predictive (NSIS-MDP), where NSIS is the Next Steps in Signaling protocol, employed for resources reservation. They also propose a general prediction technique based both on the analysis of Cell Stay Time and on the direction probabilities of hand-in and hand-out events of mobile nodes from wireless cells.

I consider this paper really interesting and relevant. I can recommend it for publication after the next few suggestions based on my expertise:

User mobility has been analyzed and a realistic 2D mobility model has been considered. They investigate different vehicular mobility models, and their impact on inter-vehicle communications. For that, they use a tool able to create urban mobility scenarios, with damaged cars and downtowns.  After that, the authors argue “There are different proposed models for describing users’ behaviors, such as [10-13]: unfortunately, all of them take into account a synthetic approach, based on analytical and/or stochastic equations, able to describe user movements in different  environments (urban, rural, etc.), without considering the existence of real topologies and road structures. In this work, instead, the Citymob4Roadmaps (C4R) proposed in [14, 15] has been considered, with realistic behaviors, because mobility traces are generated ac-60 cording to roads structures, extracted from concrete environments.” I was reviewing the list of references [10-14], but also other ones relative to driving behavior. At this point, I am missing some references to naturalistic driving method, probably the most recent advanced methodology for extracting/identifying driving behavior and driving patterns. I strongly recommend you to add some references to studies in this topic like Balsa-Barreiro et al. (2019 and 2015) focused on GIS mapping of driving behavior based on naturalistic driving data.

The authors introduce the study area “Our network consists of 65 coverage cells, with a coverage radius of about 250 meters. The considered geographical region (south of Italy) is a 3Km2 area (fig. 7), and users move according to C4R and Citymob. The APs are wired connected, by a switching subnet, to the net-sender.” Several points: (1) Why are you using a hexagonal tessellation and no other geometry –square, rectangular, etc? (2) I am missing a spatial scale in this figure/map.

Fig. 1.I am missing a spatial scale.

Fig. 8-14: Why the authors are not presenting these figures with a color legend? These present a low quality and it is difficult to differentiate lines.

Author Response

In this paper, the authors analyze the management of mobile services in Wireless Cellular Networks. In particular, they consider two classes of service: NSIS-Mobility Independent Predictive (NSIS-MIP) and NSIS-Mobility Dependent Predictive (NSIS-MDP), where NSIS is the Next Steps in Signaling protocol, employed for resources reservation. They also propose a general prediction technique based both on the analysis of Cell Stay Time and on the direction probabilities of hand-in and hand-out events of mobile nodes from wireless cells. I consider this paper really interesting and relevant. I can recommend it for publication after the next few suggestions based on my expertise:

  • User mobility has been analyzed and a realistic 2D mobility model has been considered. They investigate different vehicular mobility models, and their impact on inter-vehicle communications. For that, they use a tool able to create urban mobility scenarios, with damaged cars and downtowns.  After that, the authors argue “There are different proposed models for describing users’ behaviors, such as [10-13]: unfortunately, all of them take into account a synthetic approach, based on analytical and/or stochastic equations, able to describe user movements in different  environments (urban, rural, etc.), without considering the existence of real topologies and road structures. In this work, instead, the Citymob4Roadmaps (C4R) proposed in [14, 15] has been considered, with realistic behaviors, because mobility traces are generated according to roads structures, extracted from concrete environments.” I was reviewing the list of references [10-14], but also other ones relative to driving behavior. At this point, I am missing some references to naturalistic driving method, probably the most recent advanced methodology for extracting/identifying driving behavior and driving patterns. I strongly recommend you to add some references to studies in this topic like Balsa-Barreiro et al. (2019 and 2015) focused on GIS mapping of driving behavior based on naturalistic driving data.

We say thanks to the reviewer for his/her precious remark: as stated by the reviewer, our work differs from many others because it does not take into account mobility generated by closed equations (speed, acceleration, steering radius, etc.), but the C4R tool has been considered. It is able to generate mobility traces in which each location point belongs to real roads and follows the statistical laws of traffic. However, we provided to consider the indicated references, commenting them in section 2: after a careful reading of those paper, we are very interested at the proposed cartographic GIS idea, so we are very grateful to the reviewer, because we can now focus on the management of more concrete mobility data for our future works.     

  • The authors introduce the study area “Our network consists of 65 coverage cells, with a coverage radius of about 250 meters. The considered geographical region (south of Italy) is a 3Km2 area (fig. 7), and users move according to C4R and Citymob. The APs are wired connected, by a switching subnet, to the net-sender.” Several points: (1) Why are you using a hexagonal tessellation and no other geometry –square, rectangular, etc? (2) I am missing a spatial scale in this figure/map.

We thank the reviewer for his/her observations: (1) we provided to use hexagonal geometry in order to respect the theory of cellular systems and to consider the overlapping area between two neighboring cells (otherwise there could be some uncovered geographical regions). Of course, the shape of each coverage cell can be square, rectangular, pentagonal, etc., but we based our simulator implementation on the classical coverage theory, which provides sectorial antennas in each edge of the figure (in the case of GSM or 3G application each antenna sector covers 120 degrees) or a single base station placed at the center of the covered area, with an omni-directional antenna. In both cases, there should be an overlapping coverage portion for neighboring cells, in which the hand-over procedures should be managed (the hard, soft and seamless handover management is out of the scope of our article). As regards (2), we provided to add a clear indication of the measures of coverage radius and the sides of the illustrated map, fixing the value of the total area from 3km2 to 2.44km2.     

  • Fig. 1.I am missing a spatial scale.

We provided to add the indication of the horizontal and vertical measures.

  • Fig. 8-14: Why the authors are not presenting these figures with a color legend? These present a low quality and it is difficult to differentiate lines.

We thank the reviewer for his/her precious observation: we provided to change the color of each line, maintaining the same format for each figure.

Reviewer 3 Report

  1. I would recommend authors to add a sentence or two about the information on results obtained in their work in the abstract.
  2. for the better understanding of readers, authors should redraw figure 1, figure 2 and figure 7 with better quality and readable text.
  3. I would suggest authors should add the numbers of their results in conclusion section, list the significance of their work and future prospects too.
  4. For the benefit of readers, there should be nomenclature or abbrevations table added in the paper by authors listing all the terms use din the paper.

Author Response

  • I would recommend authors to add a sentence or two about the information on results obtained in their work in the abstract.

We thank the reviewer: we provided to add a short description at the end of the abstract of the obtained results and of the goodness of the proposed idea when compared with other literature works.

  • for the better understanding of readers, authors should redraw figure 1, figure 2 and figure 7 with better quality and readable text.

We thank the reviewer for his/her remark: we provided to zoom-in the elements of Fig. 1, adding also a reference to the measures of the left part (please refer also to the observation n. 3 of reviewer 2); Fig. 2 has been also zoomed-in for a better readability, as well as Fig. 7, with the added measures reference (please refer also to the observation n. 2 of reviewer 2). 

  • I would suggest authors should add the numbers of their results in conclusion section, list the significance of their work and future prospects too.

We are grateful to the reviewer: we provided to add, at the end of the conclusions, some of the numbers of the main results, while underlining advantages and future perspective of the proposed idea.

  • For the benefit of readers, there should be nomenclature or abbreviations table added in the paper by authors listing all the terms used in the paper

We are grateful to the reviewer for this remark, which gave us the possibility to enhance the overall quality of the paper: consequently, we provided to add a table with the abbreviations used into the text. 

Round 2

Reviewer 1 Report

My comments have been addressed with the best possible clarity. 

The redaction of the paper shall be revised. Some paragraphs in the article are too long.

I do not have any other further comments.

Author Response

We thank the reviewer for his/her opinion. We provided to restyle the paragraphs subdivision inside the paper.